# Time Reversal Symmetry for Efficient Robotic Manipulations in Deep Reinforcement Learning

**Yunpeng Jiang**
jyp9961@sjtu.edu.cn
Global College
Shanghai Jiao Tong University

**Jianshu Hu**
hjs1998@sjtu.edu.cn
Global College
Shanghai Jiao Tong University

**Paul Weng**[*]
paul.weng@dukekunshan.edu.cn
Digital Innovation Research Center
Duke Kunshan University

**Yutong Ban**[*]
yban@sjtu.edu.cn
Global College
Shanghai Jiao Tong University

## Abstract

Symmetry is pervasive in robotics and has been widely exploited to improve sample efficiency in deep reinforcement learning (DRL). However, existing approaches primarily focus on spatial symmetries—such as reflection, rotation, and translation—while largely neglecting temporal symmetries. To address this gap, we explore time reversal symmetry, a form of temporal symmetry commonly found in robotics tasks such as door opening and closing. We propose Time Reversal symmetry enhanced Deep Reinforcement Learning (TR-DRL), a framework that combines trajectory reversal augmentation and time reversal guided reward shaping to efficiently solve temporally symmetric tasks. Our method generates reversed transitions from fully reversible transitions, identified by a proposed dynamics-consistent filter, to augment the training data. For partially reversible transitions, we apply reward shaping to guide learning, according to successful trajectories from the reversed task. Extensive experiments on the Robosuite and MetaWorld benchmarks demonstrate that TR-DRL is effective in both single-task and multi-task settings, achieving higher sample efficiency and stronger final performance compared to baseline methods. Our project website and source code can be found in 1 and 2.

## 1 Introduction

Deep reinforcement learning (DRL) is a powerful machine learning framework capable of solving complex tasks, with applications across robotics, quantitative trading, and video games. Despite its successes, DRL often suffers from low sample efficiency and poor agent robustness. To address these challenges, symmetry, a common property in many real-world scenarios, has been leveraged to improve both sample efficiency and agent performance. Symmetry can be used to augment trajectories collected during training in both state-based [Lin et al., 2020, Kidziński et al., 2018] and image-based settings [Yarats et al., 2022]. Alternatively, symmetry can be embedded directly into the network architecture, making it an inherent property of the model [Cohen and Welling, 2016, Wang et al., 2022]. In addition, it can be enforced as a regularization term [Hu et al., 2024a, Raileanu et al., 2021].

---

[*]Corresponding authors.
[1]Project Page: `https://jyp9961.github.io/TR-DRL_project_page/`
[2]Source Code: `https://github.com/jyp9961/TR-DRL`

39th Conference on Neural Information Processing Systems (NeurIPS 2025).

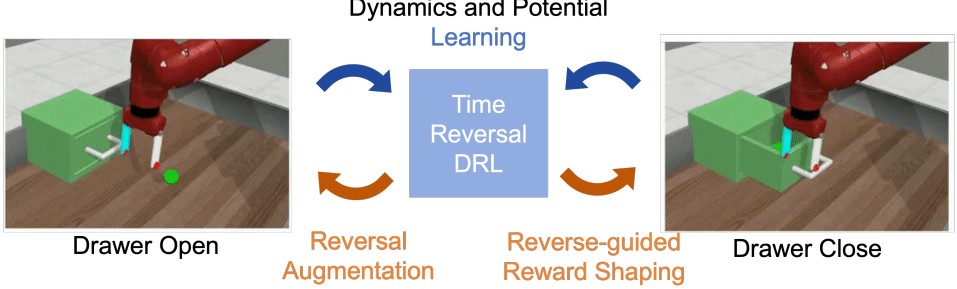

Figure 1: For a task pair, the proposed TR-DRL framework learns dynamics and potential models, leverages trajectory reversal augmentation with dynamics aware filtering and time reversal symmetry guided reward shaping, and boosts sample efficiency in both tasks.

However, existing work (see Related Work in Section 2) predominantly focuses on spatial symmetries, such as translation, reflection, and rotation, while temporal symmetries, including time-reversal symmetry and time dilation, remain largely underexplored. Intuitively, time-reversal symmetry corresponds to a reflection with respect to time, assuming actions can be reversed, which often holds in navigation tasks. Time dilation occurs in certain robotics control problems when the agent can control the speed of action execution [Hu et al., 2024b]. In this paper, we focus on leveraging time-reversal symmetry in robot manipulation tasks, where the agent controls the position and orientation of the end-effector. Unlike spatial symmetries, where augmented samples typically remain valid, temporally reversed transitions may result in invalid transitions due to complex interactions between the robot and objects.

Consider a task pair, door opening outward and door closing from outward. An augmented trajectory of closing a door from outward (Figure 2(a) from right to left) can be generated by reversing a trajectory where the agent opens the door by grasping and moving the handle outward (Figure 2(a) from left to right). In this case, the state pairs within the trajectory are fully reversible. Then consider another task pair, door opening inward and door closing from inward. When the agent closes the door by simply pushing it without grasping the handle (Figure 2(b)), reversing the trajectory becomes nontrivial. This is because the agent cannot feasibly open the door without first grasping the handle, making the reversed transitions invalid. However, certain components of the state, such as the object state (e.g., the door's opening angle), may still be reversible, even if the full transition is not. Such cases correspond to partial reversibility of the transitions where the concept of state decomposition [Pitis et al., 2020] enables isolation of dynamically reversible components.

To exploit (partial or full) time reversal symmetry in DRL, we propose a general framework (see Figure 1) that incorporates two complementary techniques, which can accelerate training for a pair of related tasks. For full time reversal symmetry, we learn an inverse dynamics model to obtain the reversed actions and generate the augmented transitions when training in both tasks. To ensure the validity of these reversed transitions, we additionally train a forward dynamics model to filter out transitions that violate the true system dynamics. For partial time reversal symmetry, the reversible component of the state can be intuitively used to guide policy learning. We leverage this form of symmetry through reward shaping, encouraging the agent for one task to follow trajectories that resemble the reversed versions of successful trajectories from the other task.

**Contributions**   Our contributions can be summarized as follows:

(i) Based on (full) time reversal symmetry (Section 3), we introduce the novel notion of partial time reversal symmetry (Section 4.1) to exploit temporal symmetry in more general settings (e.g., when objects are pushed).

(ii) We propose two techniques (Sections 4.2 and 4.3) to exploit time reversal symmetry:

- For full time reversal symmetry, transitions identified as reversible by a trained dynamics-aware filter are augmented to improve the sample efficiency of DRL algorithms.
- For partial time reversal symmetry, a reward shaping mechanism exploits transitions from successful trajectories to guide the training of the DRL agent.

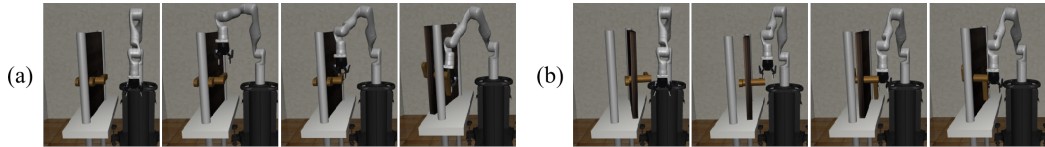

Figure 2: Examples of fully and partially reversible trajectories. (a) Fully reversible: An example of opening the door outward by grasping the handle; (b) Partially reversible: An example of closing the door from inward by pushing the door.

(iii) We conduct extensive experiments (Section 5) on standard robotics benchmarks (Robosuite, Metaworld) demonstrating that our approach significantly improves both sample efficiency and final performance compared to baseline methods. An ablation study further validates our design choices and highlights the contributions of each component within our framework.

## 2 Related Works

We summarize related works from two research directions in DRL, which include symmetry, and reward shaping techniques. For symmetry, we divide it into spatial symmetry and time reversal symmetry.

**Spatial Symmetry in DRL**    Spatial symmetry, including reflection, rotation, and translation, are extensively exploited in DRL. These symmetries enable the generation of synthetic transitions from a single environment interaction, effectively improving sample efficiency. For example, prior works [Lin et al., 2020, Corrado and Hanna, 2024, Corrado et al., 2024] have shown that applying spatial augmentations such as reflection, rotation, and translation significantly boosts sample efficiency in state-based robotics control tasks. In image-based RL, translation symmetry has also been widely adopted to enhance performance [Yarats et al., 2022, Ma et al., 2024, Hu et al., 2024a]. Data augmentation methods can also improve robustness to noise [Sinha et al., 2021, Qiao et al., 2021]. Moreover, spatial symmetry can be embedded directly into the neural network architecture through equivariance [Cohen and Welling, 2016, Wang et al., 2022, 2023], ensuring that the network respects these symmetries by design. This architectural integration reduces training time and improves generalization across diverse inputs. In contrast to these prior efforts, our work focuses on exploiting time-reversal symmetry, a form of temporal symmetry that remains underexplored in DRL.

**Time Reversal Symmetry in DRL**    Time reversal symmetry has been leveraged for data augmentation [Barkley et al., 2023] and for learning dynamics-consistent latent representations from images [Cheng et al., 2023]. In contrast to simply negating actions in reversed transitions [Barkley et al., 2023, Yao et al., 2023], our approach employs a more sophisticated strategy to derive reversed actions, making it applicable to a broader range of environments. Moreover, our method focuses on state-based control tasks, where full state information is available, eliminating the need of learning latent representations from visual observations.

Some existing works focus on exploring reversibly from goal states, utilizing the time symmetry to enhance the agent's exploration towards desired states. Starting from a goal state, the agent explores by imagining reversal steps [Edwards et al., 2018] or predicting preceding states leading to goals [Goyal et al., 2019]. Instead of using imagined trajectories, true trajectories starting from goal states are given in TRASS [Nair et al., 2020], and the agent learns from the reversed trajectories. Unlike these works, we leverage time reversal symmetry not only from goal states but for every transition in the trajectory, enabling a broader application of time symmetry across the entire state space.

Other prior works focus on enhancing the reversibility of the agent, exploring strategies to ensure that agents can backtrack or reset their actions to avoid irreversible states. For instance, Grinsztajn et al. [2021] propose to distinguish reversible from irreversible actions to improve decision-making in DRL. This distinction enables agents to prioritize reversible actions that are safer, as they guarantee the ability to backtrack if needed. Furthermore, Eysenbach et al. [2018] propose learning a reset policy alongside the normal policy to prevent agents from entering non-reversible states, ensuring safety in exploration phase and achieving better training efficiency. While their reset policy sets initial state as the ending state of the current policy and the goal state as the task's starting point, our method

treats two reversible tasks independently, with initial and goal states defined separately for each task. Additionally, our method is orthogonal to theirs and can be integrated to enhance the training of their reset policy.

**Reward Shaping in DRL**    Reward shaping is a powerful technique for enhancing the efficiency of DRL algorithms [Ibrahim et al., 2024], as it guides agents toward desired behaviors. The idea of using shaped rewards to guide learning naturally aligns with our objective of leveraging reversed trajectory in tasks of time reversal symmetry. However, reward shaping has not yet been explored in the context of time reversal symmetry. In this work, we exploit time reversal symmetry by training a potential function guided by reversed trajectories. Potential-based reward shaping [Ng et al., 1999] involves defining a potential function over the state space, which captures the agent's desired progress toward the goal. Importantly, the optimal policy remains unchanged with potential-based reward shaping, providing a theoretical foundation for its application in our method.

# 3    Background

In this section, we recall the framework of deep reinforcement learning (DRL), the soft actor-critic algorithm, the concept of time reversal symmetry, and potential-based reward shaping in reinforcement learning (RL).

**Deep Reinforcement Learning (DRL)**    For any set $\mathcal{X}$, $\Delta(\mathcal{X})$ denotes the set of probability distributions over $\mathcal{X}$. A Markov Decision Process (MDP) model $M = (\mathcal{S}, \mathcal{A}, R, T, \rho_0, \gamma)$ is defined by a set of state $\mathcal{S}$, a set of action $\mathcal{A}$, a reward function $R : \mathcal{S} \times \mathcal{A} \to \mathbb{R}$, a transition function $T : \mathcal{S} \times \mathcal{A} \to \Delta(\mathcal{S})$, a probability distribution over initial states $\rho_0 \in \Delta(\mathcal{S})$, and a discount factor $\gamma \in [0, 1]$. In RL, the agent learns a policy $\pi(\cdot \mid s) \in \Delta(\mathcal{A})$ by interacting with the environment, aiming to maximize the expected return $J = \mathbb{E}_\pi[\sum_{t=0}^\infty \gamma^t r_t \mid s_0 \sim \rho_0]$, where $\mathbb{E}_\pi$ denotes the expectation over $\pi$ and $r_t$ is the reward that the agent obtains at each timestep $t$.

**Soft Actor-Critic (SAC)**    Maximum entropy reinforcement learning (RL) addresses standard RL problems using an alternative objective that explicitly encourages stochastic policies. The objective combines cumulative reward with an entropy term: $J = \hat{\mathbb{E}}_\pi[\sum_{t=0}^\infty \gamma^t r_t + \alpha H(\pi(\cdot \mid s_t))]$, where $\gamma$ is the discount factor, $\alpha$ is a trainable coefficient of the entropy term, and $H(\pi(\cdot \mid s_t))$ represents the entropy of of the policy distribution $\pi(\cdot \mid s_t)$. The Soft Actor-Critic (SAC) algorithm [Haarnoja et al., 2018] optimizes this objective by training the actor $\pi_\theta$ and critic $Q_\psi$ with the following losses:

$$
\begin{aligned}
L_\pi(\theta) &= \hat{\mathbb{E}}_{s_t \sim \mathcal{D}, a \sim \pi}[\alpha \log \pi_\theta(a \mid s_t) - Q_\psi(s_t, a)], \\
L_Q(\psi) &= \hat{\mathbb{E}}_{s_t, a_t \sim \mathcal{D}}[(Q_\psi(s_t, a_t) - \hat{Q}(s_t, a_t))^2],
\end{aligned}
\tag{1}
$$

where $\hat{Q}(s_t, a_t) = r_t + \gamma Q_{\bar{\psi}}(s_{t+1}, a_{t+1}) - \alpha \log \pi_\theta(a_{t+1}|s_{t+1})$, which is the target Q-value computed using a target network, and $a_{t+1} \sim \pi_\theta(\cdot \mid s_{t+1})$. Here, $\theta$, $\psi$ and $\bar{\psi}$ represent the parameters of the actor, the critic and the target critic respectively, while $\mathcal{D}$ represents the replay buffer. To stabilize training, the weights of the target network are updated as an exponential moving average of the online critic network's weights.

**Time Reversal Symmetry in DRL**    Given an involution[2] $f : \mathcal{S} \times \mathcal{A} \times \mathcal{S} \to \mathcal{S} \times \mathcal{A} \times \mathcal{S}$, an MDP satisfies (full) time reversal symmetry (adapted from Barkley et al. [2023]) if for all $s_t, s_{t+1} \in \mathcal{S}$,

$$
T(s_{t+1} \mid s_t, a_t) = T(\breve{s}_t \mid \breve{s}_{t+1}, \breve{a}_t).
\tag{2}
$$

where $(\breve{s}_{t+1}, \breve{a}_t, \breve{s}_t) = f(s_t, a_t, s_{t+1})$ and $\breve{\cdot}$ denotes time reversal operation on $\mathcal{S}$ or $\mathcal{A}$. Intuitively, involution $f$ represents the symmetry that reverses the passage of time. Note that in some situations, it can be simply written as $f(s, a, s') = (f_\mathcal{S}(s), f_\mathcal{A}(a), f_\mathcal{S}(s'))$ using an involution $f_\mathcal{S}$ over states and an involution $f_\mathcal{A}$ over actions. An example of time reversal symmetry in physical system is the transformation of position $p$, momentum $q$, and the applied force $a$. The involution $f_\mathcal{S}$ transforms state $s = (q, p)$ into $f_\mathcal{S}(s) = (q, -p)$, preserving position while negating momentum, which is a common phenomenon in physical systems. For the action $a$, $f_\mathcal{A}$ reverses the applied force such that $f_\mathcal{A}(a) = -a$. This ensures that the dynamics remain consistent under time reversal.

---

[2] Recall an involution is a one-to-one mapping, which is its own inverse.

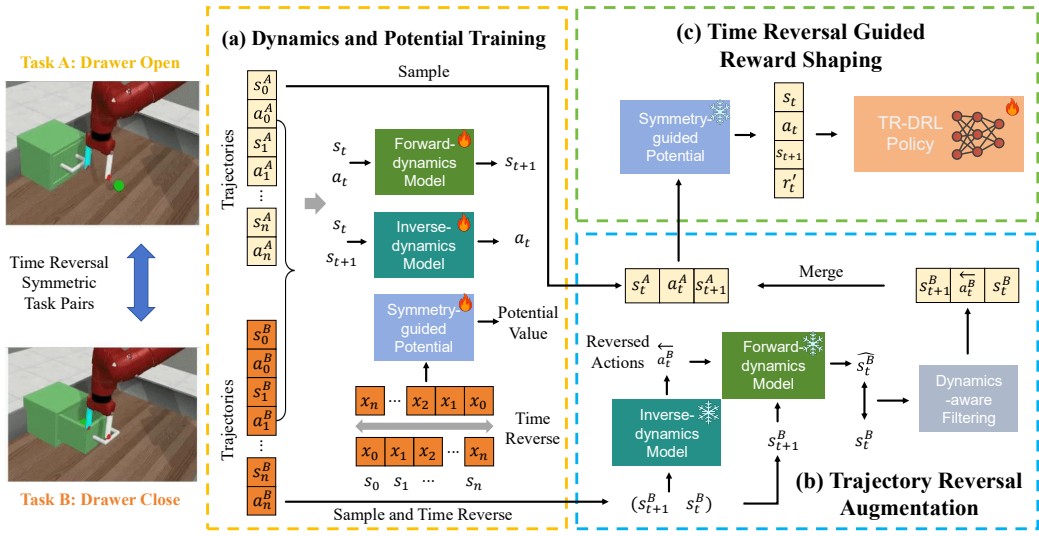

Figure 3: **Overview of our TR-DRL**. We learn dynamics and potential models, apply reversal augmentation on transitions from the reversed task, and apply time reversal symmetry guided reward shaping on all transitions.

**Potential-Based Reward Shaping** We recall the concept of potential-based reward shaping proposed by Ng et al. [1999]. A shaping reward function $\mathcal{F} : \mathcal{S} \times \mathcal{A} \times \mathcal{S} \to \mathbb{R}$ is potential-based if there exists a real-valued function $\Phi : \mathcal{S} \to \mathbb{R}$ such that for all $s \in \mathcal{S}$, $a \in \mathcal{A}$, $s' \in \mathcal{S}$,

$$\mathcal{F}(s, a, s') = \gamma\Phi(s') - \Phi(s), \tag{3}$$

This condition is necessary and sufficient to ensure that an optimal policy of the modified MDP $M' = (\mathcal{S}, \mathcal{A}, R + \mathcal{F}, T, \rho_0, \gamma)$ remains optimal in the original MDP $M = (\mathcal{S}, \mathcal{A}, R, T, \rho_0, \gamma)$.

## 4   Methodology

In this section, we first define the problem set-up considered in this paper and introduce two types of time-reversal symmetry, full or partial time reversal symmetries, for which we provide illustrative examples in robotics (Section 4.1). We then propose a generic method, as shown in Figure 3, which applies trajectory reversal augmentation (Section 4.2) on fully reversible transitions identified by our proposed dynamics-consistent filter, and employs reward shaping (Section 4.3) guided by partially reversible transitions.

### 4.1   Problem Formulation

In this paper, we assume that the RL agent aims at learning to solve (at least) two related tasks in the same environment (e.g., door opening/closing or peg insertion/removal). For such a pair of tasks, the RL agent may learn in a more data-efficient way by exploiting full time reversal (FTR) symmetry and partial time reversal (PTR) symmetry (see definition below). While Barkley et al. [2023] assume that the FTR symmetry holds globally and that the involution to transform actions is known, which makes this temporal symmetry property too restrictive and difficult to apply in practice, we do not make these two assumptions, which allows us to consider scenarios like the next example. However, the challenge now is to detect when Equation (2) holds and learn to recover reverse actions $\bar{a}_t$.

**Example 1.** Consider a pair of manipulation tasks: peg insertion and peg removal. Assume end-effector position control and that the state includes the positions of both the end-effector and the object (i.e., peg). When the robot arm holds and moves the peg towards the hole, for a transition $(s_t, a_t, s_{t+1})$, reversing the action enables the agent to move from $s_{t+1}$ back to $s_t$ without violating the true dynamics. This means that all transitions along this trajectory exhibit FTR symmetry. However, contacts and frictions prevent the definition of involution $f$ and if the peg can be dropped, the transitions are naturally not reversible anymore.

While the relaxation of these two assumptions extend the applicability of FTR symmetry, for many pairs of tasks, an even weaker notion of temporal symmetry may be needed. We therefore introduce the novel notion of partial time reversal (PTR) symmetry:

**Partial Time Reversal (PTR) Symmetry**   Assume that a state $s \in \mathcal{S}$ (resp. $s' \in \mathcal{S}$) can be decomposed into two parts $(x, y) \in \mathcal{X} \times \mathcal{Y}$ (resp. $(x', y') \in \mathcal{X} \times \mathcal{Y}$) and that an involution $f_\mathcal{X} : \mathcal{X} \to \mathcal{X}$ is given.   A pair of states $(s, s') \in \mathcal{S}^2$ satisfies PTR symmetry if there exist $(\bar{y}, \bar{y}') \in \mathcal{Y}^2$ and $(a, \bar{a}) \in \mathcal{A}^2$ such that:

$$T(s' \mid s, a) = T(\bar{s} \mid \bar{s}', \bar{a}), \tag{4}$$

where $\bar{x} = f_\mathcal{X}(x)$, $\bar{x}' = f_\mathcal{X}(x')$, $\bar{s} = (\bar{x}, \bar{y})$, and $\bar{s}' = (\bar{x}', \bar{y}')$. Intuitively, $\mathcal{X}$ is the part that is reversible (e.g., containing object state information). Using this weaker property, we can now account for scenarios like the following example:

**Example 2.**   Consider another pair of tasks: door opening and door closing inward, with a similar definition of state and action spaces as in Example 1. In the door closing task, the agent learns to close the door by pushing it, without grasping the handle. Along this trajectory, the transitions do not satisfy FTR symmetry, as there does not exist an action that allows the robot arm to pull the door without grasping the handle. However, the object (i.e. door) state remains reversible. We can find corresponding state pairs with reversed object state in the trajectories of door opening tasks. These pairs reflect PTR symmetry, as only the object component of the state is reversible.

In the next two subsections, we explain how to exploit FTR and PTR symmetries in DRL.

## 4.2   Trajectory Reversal Augmentation with Dynamics-Aware Filtering

In this section, we introduce how we augment the fully reversible transitions and how these fully reversible transitions are detected by a dynamics-consistent filter. Given a pair of tasks with time reversal symmetry, any transition $(s, a, s')$ exhibiting FTR symmetry defined above can be augmented by generating its reversed transition $(s', \bar{a}, s)$ and incorporating it into DRL training. Now the problem to be solved is finding $\bar{a}$. In some robotics tasks, a straightforward choice of $\bar{a}$ is to negate the action which corresponds to reversing forces or torques, i.e. $\bar{a} = -a$ [Barkley et al., 2023]. However, it does not work for tasks involving contact dynamics or non-linear effects. To address this, we propose a more general approach by learning an inverse dynamics model $h$, represented by a neural network:

$$a = h(s, s'), \tag{5}$$

which is trained using transitions collected during RL training by minimizing the following loss:

$$L_h = \hat{\mathbb{E}}_{(s,a,s') \sim \mathcal{D}}[(h(s, s') - a)^2], \tag{6}$$

where $\hat{\mathbb{E}}$ is an empirical mean estimating the expectation over the true data distribution and $\mathcal{D}$ denotes the replay buffer containing transitions $(s, a, s')$. Since the pair of reversible tasks share the same underlying dynamics, a single inverse dynamics model can be trained jointly using transitions from both tasks. This shared model ensures the accurate inverse predictions when applied to the reversed task.

Note that trajectory reversal augmentation can only be applied directly on fully reversible transitions. To identify such transitions, an additional dynamics-consistent filter is introduced to select appropriate samples from the replay buffer. This filtering is achieved by training a forward dynamics model $g$ on transitions from both tasks by minimizing the following loss:

$$L_g = \hat{\mathbb{E}}_{(s,a,s') \sim \mathcal{D}}[(g(s, a) - s')^2]. \tag{7}$$

This model allows us to verify whether a reversed transition $(s', \bar{a}, s)$ is consistent with the underlying dynamics. In particular, for a reversed transition $(s', \bar{a}, s)$, we feed the state $s'$ and action $\bar{a}$ into the forward dynamics model $g$ to get the predicted state $\hat{s}$:

$$\hat{s} = g(s', \bar{a}) = g(s', h(s', s)). \tag{8}$$

The error between the predicted state $\hat{s}$ and the true state $s$ serves as a measure of feasibility. Only when this prediction error $\| s - \hat{s} \|$ is below a predefined threshold $\beta$, the reversed transition is considered valid and included in the training for the reversed task.

## 4.3 Time Reversal Symmetry Guided Reward Shaping

In scenarios where not all transitions are fully reversible, trajectory reversal augmentation may become less effective. As an illustration, consider the task mentioned in Example 2. In such cases, most reversed transitions are filtered out by the dynamics-consistent filter, since the agent cannot reverse the action (i.e., from "push the door" to "pull the door") without first grasping the handle. However, we can still exploit partial time reversal symmetry to improve sample efficiency. In many tasks, the object state, such as the position of a door or the placement of a peg, remains reversible, while irreversibility arises primarily from the agent state, such as joint angles or gripper force.

To exploit this separation, we examine the relationship between object states in the trajectories of a pair of partially reversible tasks. Let $x_t$ denote the object-related component of the full state $s_t$ at time step $t$. Given a high-reward trajectory $\tau = (s_0, s_1, ..., s_n)$ from one task, another trajectory $\bar{\tau} = (\bar{s}_0, \bar{s}_1, ..., \bar{s}_n)$ from the reversed task should likewise receive a high reward if it achieves the reversed sequence $(x_n, x_{n-1}, ..., x_0)$, or a partial reward if it accomplishes only a portion of the reversed sequence. This observation raises a key question: can we leverage the partially reversible time symmetry, such that the reversible object-related components can be used to accelerate agent training?

As an answer to this question, we propose time reversal symmetry guided reward shaping. Here we employ potential-based reward shaping [Ng et al., 1999] since it preserves policy optimality and directly operates on states. To fully utilize multiple successful trajectories, we propose to train a potential model $\Phi$ for the reversed task, which maps the object state $x_t$ to a potential value $\Phi(x_t)$. As discussed, for the reversed trajectory containing sequences from $x_t$ to $x_0$, the potential values of these object states should increase in the reversed task. Therefore, the object states along this reversed trajectory are labeled with potential values ranging from $0$ to $1$, and used to train the reversed task. Here, a linear function can be used to interpolate potential values between the start and end states. The potential model $\Phi$, is then trained to minimize the following loss:

$$
L_\Phi = \hat{\mathbb{E}}_{\tau=(s_0,s_1,...,s_n)\sim\mathcal{B}} \left[ \left( \Phi(x_t) - \frac{n-t}{n} \right)^2 \right],
$$

$$
s_t = [x_t, y_t], t \in (0, ..., n)
$$

(9)

where $\mathcal{B}$ denotes the dataset which includes high-reward trajectories.

Based on Equation (3), the reward for each transition $(\bar{s}_t, \bar{a}_t, \bar{s}_{t+1}, \bar{r}_t)$ in the reversed task is reshaped as $\bar{r}_t + \gamma\Phi(x_{t+1}) - \Phi(x_t)$ during training. This potential-based reward shaping mechanism encourages the agent to align the trajectory of object states with the reversed successful trajectory by dynamically shaping rewards based on the potential values. In the door tasks, for example, a closed-to-open trajectory, reversed from a successful open-to-closed trajectory, guides the door opening agent by training $\Phi$ to predict low potential values for closed door states and high potential values for open door states. Since potential-based reward shaping assigns a distinct potential value to each object state within the trajectory, reflecting its proximity to the success state, this smooth progression of potential values improves agent training by offering step-by-step guidance towards the goal state.

## 4.4 TR-DRL Algorithm

Our proposed techniques to exploit time reversal symmetry can be integrated in various DRL algorithms. For concreteness, an example with SAC is presented in Algorithm 1. The training process alternates between two reversible tasks. First, the agents collect data from their respective environments. Then transitions from both environments are used to train the forward and inverse dynamics models. Meanwhile, successful trajectories are employed to update the potential models. During agent training, we augment the original samples from the agent's current task with reversed samples from the reversible task via trajectory reversal augmentation and dynamics-aware filtering. Further, we apply time reversal symmetry guided reward shaping to reshape rewards of all the transitions. Finally, we update the agent with DRL loss.

---

**Algorithm 1** TR-DRL

---

**Required**: a pair of reversible tasks $(A, B)$, total number of training episodes $N$, total number of timesteps in one episode $T$.

1: Initialize empty replay buffers $\mathcal{D}_\mathcal{A}$ and $\mathcal{D}_\mathcal{B}$. Initialize actor $\pi$ and critic $Q$.
2: Initialize potential models. Initialize forward and inverse dynamics models.
3: **for** $n = 0 \ldots N$ **do**
4:     **for** $t = 0 \ldots T$ **do**
5:         Alternate the following training steps between $A$ and $B$.
6:         // Task $A$:
7:         The agent interacts with the environment and save the transition in replay buffer $\mathcal{D}_A$.
8:         Update forward and inverse dynamics models using Equation (6) and Equation (7).
9:         Update potential models using Equation (9).
10:         Sample two minibatches $d_A$ and $d_B$ from $\mathcal{D}_\mathcal{A}$ and $\mathcal{D}_\mathcal{B}$.
11:         Generate $d_{B,aug}$ from $d_B$ by reversal augmentation with dynamics-aware filtering.
12:         Apply time reversal symmetry guided reward shaping on $d_A \cup d_{B,aug}$.
13:         Update the actor and critic, $\pi$ and $Q$, with $d_A \cup d_{B,aug}$ using Equation (1).
14:         // Task $B$: ...
15:     **end for**
16: **end for**

---

## 5 Experimental Results

To demonstrate the effectiveness of our proposed method, we conduct comprehensive experiments to assess the performance of our approach in both single-task and multi-task settings. We also run ablation study on our method to illustrate the design choice of different components.

**Experimental setup** To validate our method, we evaluate our method in 60 environments from two standard robotics control benchmarks, Meta-World [Yu et al., 2020] and Robosuite [Zhu et al., 2025]. Detail introductions and example figures of these environments are provided in Section A. We use SAC [Haarnoja et al., 2018], multi-task SAC [Yu et al., 2020], and multi-headed SAC [Yu et al., 2020] as the baselines for comparison. Hyperparameters, such as network architecture and learning rates, are listed in Section B. We use sparse rewards in all our experiments, which makes learning challenging for the agent. To mitigate this, we initialize the agent's replay buffer with 10 expert demonstration trajectories, providing guidance to agent's exploration. Unless specified, all reported scores are averaged over five runs, with standard deviations included in the results. Throughout each run, the agent is evaluated every 20 training episodes by calculating the average success rate of 20 evaluation episodes. To present the aggregated performance, we compute the inter-quartile mean (IQM) as proposed by Agarwal et al. [2021].

### 5.1 Main results

**Robosuite-Single task** To demonstrate the efficiency of our proposed method, we first evaluate it under single-task setting in Robosuite, where we train an agent for each task. Here, five pairs of tasks exhibiting time reversal symmetry are considered: door opening/closing inward, door opening/closing outward, peg insertion/removal, nut assembly/disassembly and block stacking/unstacking. The IQM of agent performance across 10 environments are shown in Figure 4, and full evaluation curves are provided in Figure 24 due to page limit. The results demonstrate the performance gain of our method over the baseline and confirm the contribution of each component in our method.

**Robosuite-Multi task** We further evaluate our method in multi-task settings. Our method is orthogonal to existing multi-task learning frameworks, meaning it can be seamlessly integrated with them. To highlight the performance gains of our approach, we demonstrate its effectiveness by combining it with existing multi-task methods. Here, we start with training one agent for a pair of tasks. Later on, we extend our method to using a single agent for all concerned tasks. To train a single agent for multiple tasks, we consider extend the models to either taking an additional task embedding as input (task-conditioned) or outputting the actions of several tasks at the same time (multi-head).

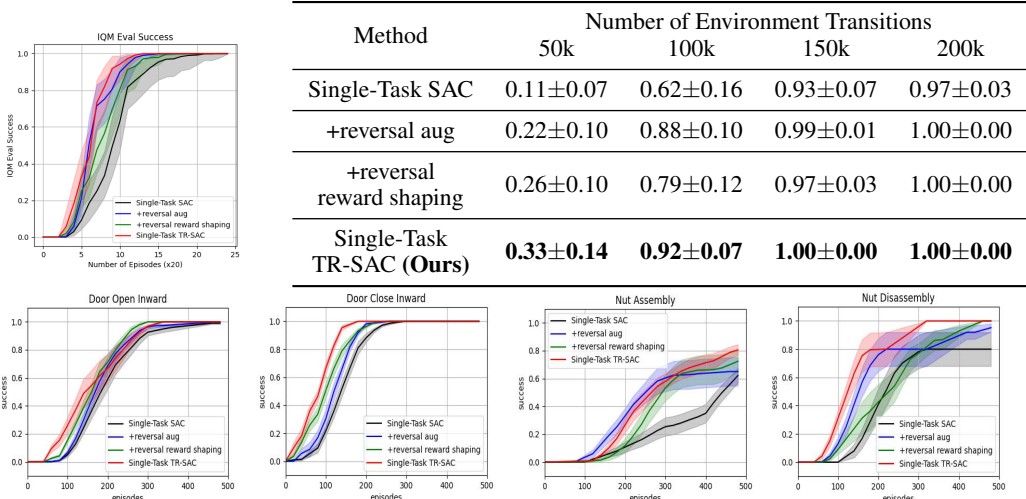

| Method | Number of Environment Transitions | | | |
| --- | --- | --- | --- | --- |
| | 50k | 100k | 150k | 200k |
| Single-Task SAC | 0.11±0.07 | 0.62±0.16 | 0.93±0.07 | 0.97±0.03 |
| +reversal aug | 0.22±0.10 | 0.88±0.10 | 0.99±0.01 | 1.00±0.00 |
| +reversal reward shaping | 0.26±0.10 | 0.79±0.12 | 0.97±0.03 | 1.00±0.00 |
| Single-Task TR-SAC **(Ours)** | **0.33±0.14** | **0.92±0.07** | **1.00±0.00** | **1.00±0.00** |

Figure 4: **Results for single-task setting in 10 environments from Robosuite.** Top: Plots and table for IQM of success rate. Bottom: Curves of success rate in two pair of reversible tasks. "Single-Task SAC": baseline; "+reversal aug": trajectory reversal augmentation with dynamics-aware filtering; "+reversal reward shaping": time reversal symmetry guided reward shaping.

For task-conditioned setting with only two tasks, we use one-hot encoding for the task embedding. The actor, critic and potential models take both the state and the task embedding as input. Considering that the environment dynamics are identical within each task pair, the pair of tasks share the forward and inverse dynamics models. For multi-headed setting with only two tasks, the models output values for both two tasks simultaneously. The performance of integrating our proposed method into these baselines in 10 environments of Robosuite is shown in Figure 5. The full evaluation curves of agent performance are included in Figure 25. Our proposed techniques clearly enhance the sample efficiency and improve the final performance when combined with the three baselines.

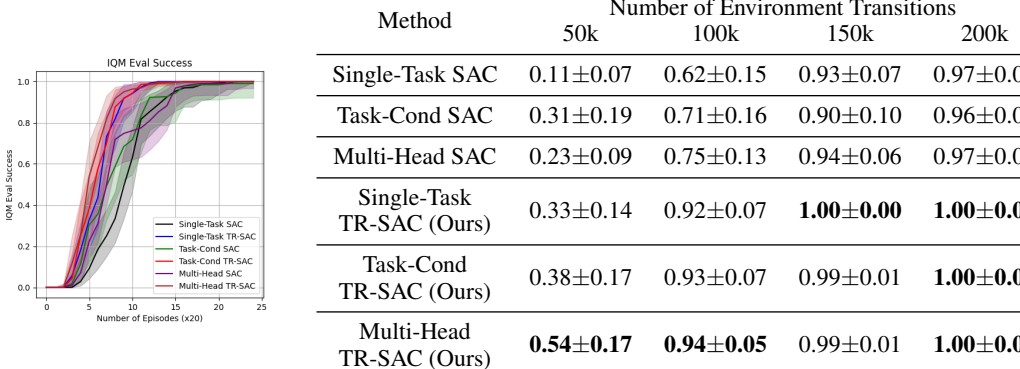

| Method | Number of Environment Transitions | | | |
| --- | --- | --- | --- | --- |
| | 50k | 100k | 150k | 200k |
| Single-Task SAC | 0.11±0.07 | 0.62±0.15 | 0.93±0.07 | 0.97±0.03 |
| Task-Cond SAC | 0.31±0.19 | 0.71±0.16 | 0.90±0.10 | 0.96±0.04 |
| Multi-Head SAC | 0.23±0.09 | 0.75±0.13 | 0.94±0.06 | 0.97±0.03 |
| Single-Task TR-SAC (Ours) | 0.33±0.14 | 0.92±0.07 | **1.00±0.00** | **1.00±0.00** |
| Task-Cond TR-SAC (Ours) | 0.38±0.17 | 0.93±0.07 | 0.99±0.01 | **1.00±0.00** |
| Multi-Head TR-SAC (Ours) | **0.54±0.17** | **0.94±0.05** | 0.99±0.01 | **1.00±0.00** |

Figure 5: **IQM of success rate for multi-task settings in 10 environments from Robosuite.** "Task-Cond" and "Multi-Head" are short for "task-conditioned" and "multi-headed" respectively.

**Metaworld-Multi task** Furthermore, we evaluate our methods on MT50, a benchmark with 50 environments from Meta-World. Within the 50 tasks, we identify 12 pairs of reversible taks and apply our techniques to these pairs. Here, considering the exploding output dimensions when using multi-head setting for 50 tasks, we remove this baseline. Instead, we introduce another baseline called language-conditioned SAC. Here, the task embeddings are obtained by applying a pretrained language encoder, called CLIP [Radford et al., 2021], on the language instructions of these tasks. The IQM results for these 12 task pairs are shown in the right of Figure 6, and additional results including the average number of training episodes required to achieve a 100% success rate are included in Table 2 and Figure 26. We also present results for all 50 environments of MT50 in Figure 27 and

Figure 28. With our proposed techniques, the agent learns faster and performs better compared to the baselines in both reversible tasks and all tasks of MT50.

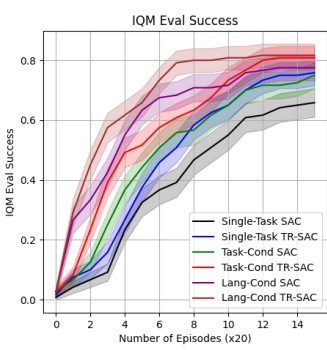

| Method | Number of Environment Transitions | | |
| --- | --- | --- | --- |
| | 50k | 100k | 150k |
| Single-Task SAC | 0.33±0.05 | 0.55±0.05 | 0.66±0.05 |
| Task-Cond SAC | 0.44±0.05 | 0.65±0.05 | 0.75±0.04 |
| Lang-Cond SAC | 0.63±0.05 | 0.72±0.05 | 0.78±0.04 |
| Single-Task TR-SAC (Ours) | 0.38±0.05 | 0.65±0.05 | 0.76±0.04 |
| Task-Cond TR-SAC (Ours) | 0.52±0.05 | 0.73±0.04 | 0.81±0.04 |
| Lang-Cond TR-SAC (Ours) | **0.66±0.05** | **0.81±0.04** | **0.82±0.04** |

Figure 6: **IQM of success rate for multi-task settings in 12 pair of reversible tasks in MT50 of Meta-World.** "Task-Cond" and "Lang-Cond" are short for "task-conditioned" and "language-conditioned" respectively.

## 5.2  Ablation study

**Trajectory reversal augmentation with dynamics-aware filtering**  We analyze trajectory reversal augmentation with dynamics-aware filtering on three pairs of tasks: door opening/closing inward, door opening/closing outward, and peg insertion/removal. As shown in Section C, incorporating reversed transitions improves the performance for fully reversible tasks (e.g., door opening/closing outward and peg insertion/removal). However, most transitions in door opening/closing inward are not fully reversible. Including the reversed transitions generated by the inverse dynamics model leads to infeasible transitions, resulting in degraded performance. After incorporating dynamics-aware filtering, which removes invalid reversed transitions, the performance surpasses the baseline for partially reversible tasks, demonstrating the effectiveness of our filtering strategy. As for the hyperparameter $\beta$ that controls the filtering error tolerance, we finalize its value as $0.01$ after tuning among $[0.01, 0.001, 0.0001]$, with the related results presented in Section D.

**Time reversal symmetry guided reward shaping**  Here we investigate the design choice for the time reversal symmetry-guided reward shaping. We first explore how the potential models should be trained. As shown in Section F, it is concluded that two potential models should be trained with successful trajectories from the task itself, and from its reversible counterpart respectively. Under this setting, the average of the rewards from these two models are used as the final reward. Moreover, four different types of potential value functions along the successful trajectories are compared. The linear function outperforms the other choices, as shown in Section G.

## 6  Conclusion

We propose TR-DRL, a framework leveraging time reversal symmetry to enhance sample efficiency of DRL algorithms. Key contributions include a novel notion of partial time reversal symmetry, trajectory reversal augmentation with dynamics-aware filtering, and symmetry-guided reward shaping. Experiments on Robosuite and Metaworld demonstrate improved agent performance and learning efficiency. Future work may explore using prediction errors to identify reversible task pairs automatically, which allows deep reinforcement learning in robotics to be more efficient.

## Acknowledgments

This work has been supported by the program of National Natural Science Foundation of China (No. 62176154), by the program of National Natural Science Foundation of China (No. 6250020129), and by Shanghai Magnolia Funding Pujiang Program (No. 23PJ1404400).

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

Figure 7: Door closing inward.

Figure 8: Door opening inward.

Figure 9: Door closing outward.

Figure 10: Door opening outward.

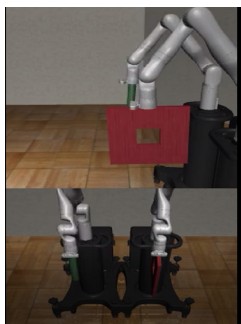 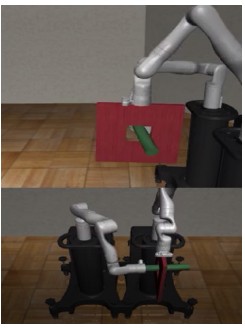 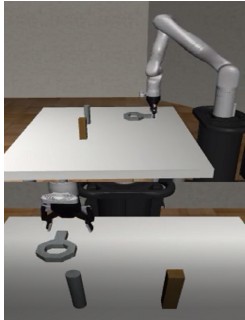 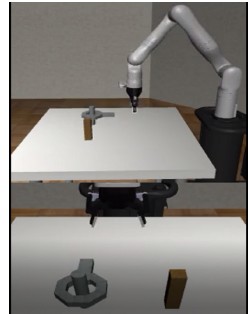

Figure 11: Peg insertion.

Figure 12: Peg removal.

Figure 13: Nut assembly.

Figure 14: Nut disassembly.

## A Environments

We introduce the environments utilized in our Robosuite experiments.

- Door Opening/Closing Inward: The agent needs to open/close the door inward. "Inward" means that the door is on the same side as the robotics arm. The agent can close the door by pushing it. To open the door, the agent has to grasp the handle and pull the handle to a desired position, making this task pair partially time reversal symmetric. Examples are shown in Figure 7 and Figure 8.

- Door Opening/Closing Outward: The agent needs to open/close the door outward. "Outward" indicates that the door is on the opposite side as the robotics arm. In this task pair, the agent has to grasp the handle and then open/close the door, making this task pair fully time reversal symmetric. Examples are shown in Figure 9 and Figure 10.

- Peg Insertion/Removal: The agent needs to insert/remove the peg into/out of the hole. Examples are shown in Figure 11 and Figure 12.

- Nut Assembly/Disassembly: The agent needs to assemble/disassemble the nut. Examples are shown in Figure 13 and Figure 14.

- Block Stack/Unstack. The agent needs to either stack a small block onto a larger one or unstack it by removing the small block.

The introductions of environments that we have used in our Meta-World experiments can be found in [Yu et al., 2020], which also provides the language instruction for each task.

## B Implementation Details

As shown in Table 1, we present the value of hyperparameters used in our experiments. For experiments in Robosuite, we adopt the hyperparameters specified by Haarnoja et al. [2018]. In

the case of experiments on MT50 of Metaworld, we primarily follow the hyperparameter settings provided by Yu et al. [2020]. Furthermore, in our source code 2, we include the implementation of our method built upon MOORE [Hendawy et al., 2024], a more recent and advanced baseline compared to SAC for multi-task RL.

| Hyperparameter | Robosuite | MetaWorld |
|---|---|---|
| hidden depth | 2 | 3 |
| hidden dimension | 512 | 400 |
| horizon | 500 | 200 |
| environment steps | 250,000 | 100,000 |
| replay buffer capacity | 250,000 | 100,000 |
| random steps | 5,000 | 2,000 |
| batch size | 512 | 128 |
| discount | 0.99 | 0.99 |
| learning rate | 1e-3 | 3e-4 |
| learning rate ($\alpha$ of SAC) | 1e-3 | 3e-4 |
| target network update frequency | 2 | 1 |
| target network soft-update rate | 0.01 | 0.005 |
| actor update frequency | 2 | 2 |
| actor log stddev bounds | [-10, 2] | [-20, 2] |
| init temperature | 0.1 | 0.1 |

Table 1: Hyperparameters used in our experiments.

## C  Ablation Study of Dynamics-Aware Filtering in Trajectory Reversal Augmentation

As shown in Figure 15, incorporating reversed transitions improves the performance for fully reversible tasks (e.g., door opening/closing outward and peg insertion/removal). However, most transitions in door opening/closing inward are not fully reversible. Including the reversed transitions generated by the inverse dynamics model leads to infeasible transitions, resulting in degraded performance. After incorporating dynamics-aware filtering, which removes invalid reversed transitions, the performance surpasses the baseline for partially reversible tasks, demonstrating the effectiveness of our filtering strategy.

## D  Hyperparameter Tuning in Dynamics-Aware Filtering

Here, we perform hyperparameter tuning of $\beta$ in dynamics-aware filtering. Recall that $\beta$ governs the error tolerance for reversed transitions: $\beta = 0.01$ filters out transitions where $\| s - \hat{s} \| \geq 0.01 \cdot \| s_{\max} - s_{\min} \|$, where $s_{\max}$ and $s_{\min}$ represent the state space extremes. We conduct a linear search for $\beta$ among $[0.01, 0.001, 0.0001]$. Evaluation curves of agent success rate in these six environments are shown in Figure 16, while the inter-quartile mean of agent success rate is presented in Figure 17. Based on these results, we select $\beta = 0.01$ for subsequent experiments.

## E  Separate Plots of Figure 5 and Figure 6

Due to the page limit in the main text, we combine the results of all methods into a single plot for Figure 5 and Figure 6. As shown in Figure 18 and Figure 19, we include separate plots for each algorithm pair (TR vs. no TR) for better readability.

## F  Ablation Study of Potential Models

In our ablation study of potential models, we evaluated four training strategies: (1) using only the task's own successful trajectories to train one potential model, (2) using only the reversible task's trajectories to train one potential model, (3) training a joint potential model with successful trajectories

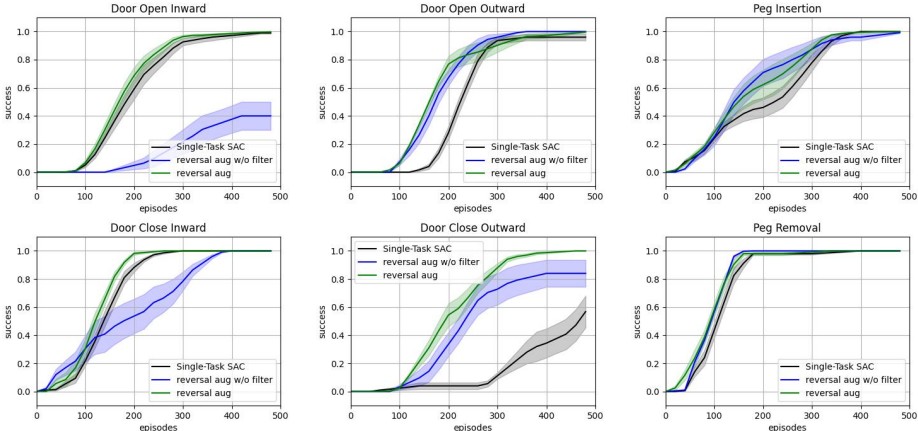

Figure 15: Evaluation curves of trajectory reversal augmentation with dynamics-aware filtering in 6 environments of robosuite. "Single-Task SAC" serves as the baseline. "+reversal aug w/o filter" introduces trajectory reversal augmentation without filtering, while "+reversal aug" incorporates dynamics-aware filtering for reversed transitions.

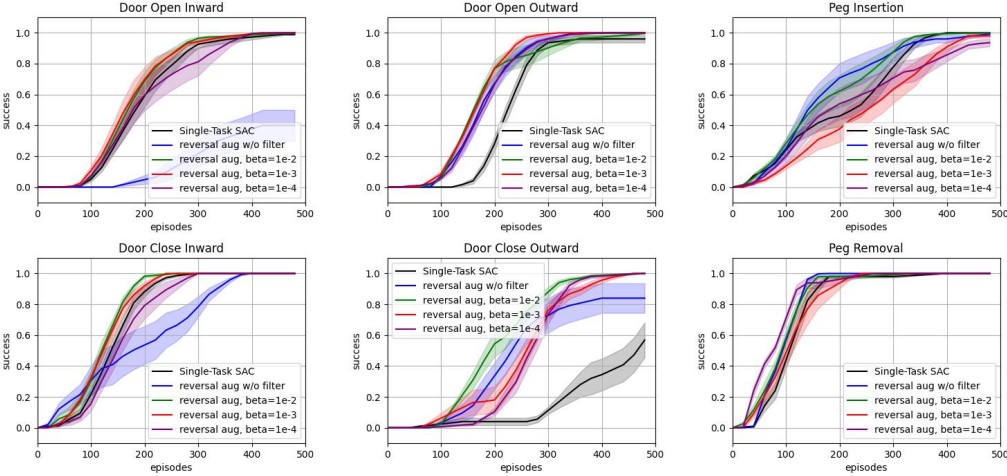

Figure 16: Evaluation curves of agent success rate using trajectory reversal augmentation with dynamics-aware filtering (different $\beta$).

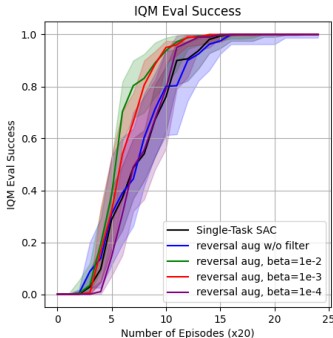

Figure 17: IQM of agent success rate using trajectory reversal augmentation with dynamics-aware filtering with different $\beta$.

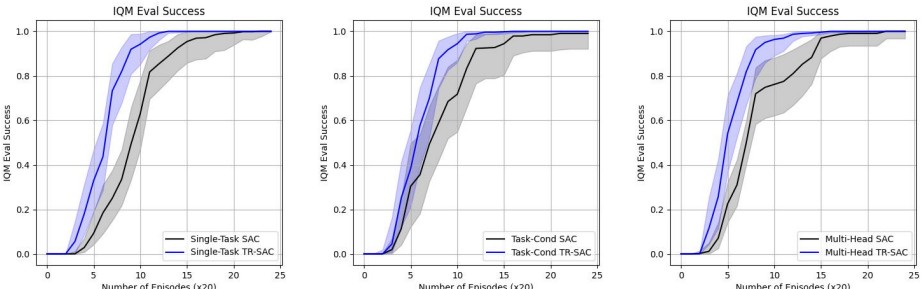

Figure 18: IQM of success rate for multi-task settings in 10 environments from Robosuite with separate plots for each algorithm pair (TR vs. no TR). "Task-Cond" and "Multi-Head" are short for "task-conditioned" and "multi-headed" respectively.

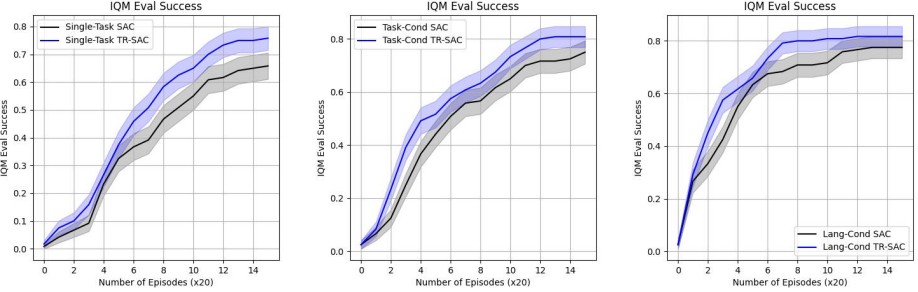

Figure 19: IQM of success rate for multi-task settings in 12 pair of reversible tasks in MT50 of Meta-World with separate plots for each algorithm pair (TR vs. no TR). "Task-Cond" and "Lang-Cond" are short for "task-conditioned" and "language-conditioned" respectively.

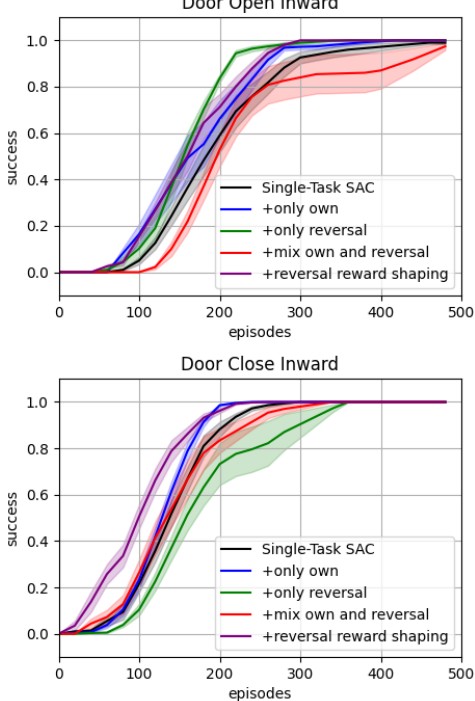

Figure 20: Ablation study of potential models in the task pair of door opening/closing inward. "Single-Task SAC" serves as the baseline. "+only own" indicates using only the task's own successful trajectories to train one potential model. "only reversal" indicates using only the reversible task's trajectories to train one potential model. "mix own and reversal" indicates training a joint potential model with successful trajectories from two tasks. "+reversal reward shaping" indicates training two separate potential models, one for successful trajectories from its own and the other for successful trajectories from the reversible task.

from two tasks, and (4) training two separate potential models, one for successful trajectories from its own and the other for successful trajectories from the reversible task. The final reward value is then computed as the average of the rewards obtained from these two models. Based on the ablation study shown in Figure 20, we conclude that training two potential models is always better than the baseline. Therefore, we use this setting to train potential models in subsequent experiments.

## G    Ablation Study of Potential Value Labeling Function in Potential-Based Reward Shaping

We evaluate four monotonically increasing functions as the potential value labeling function for a successful trajectory of length $n$.

- Linear: $\Phi(s_t) = \frac{t}{n}$,

- Triangular: $\Phi(s_t) = \frac{t(t+1)}{n(n+1)}$,

- Original Geometric: $\Phi(s_t) = \gamma^{n-t}$,

- Geometric: $\Phi(s_t) = \frac{\gamma^{n-t} - \gamma^{n-1}}{1 - \gamma^{n-1}}$.

Results are shown in Figure 21, with inter-quartile mean (IQM) in Figure 22. Based on these results, we adopt the linear function for potential-based reward shaping in subsequent experiments.

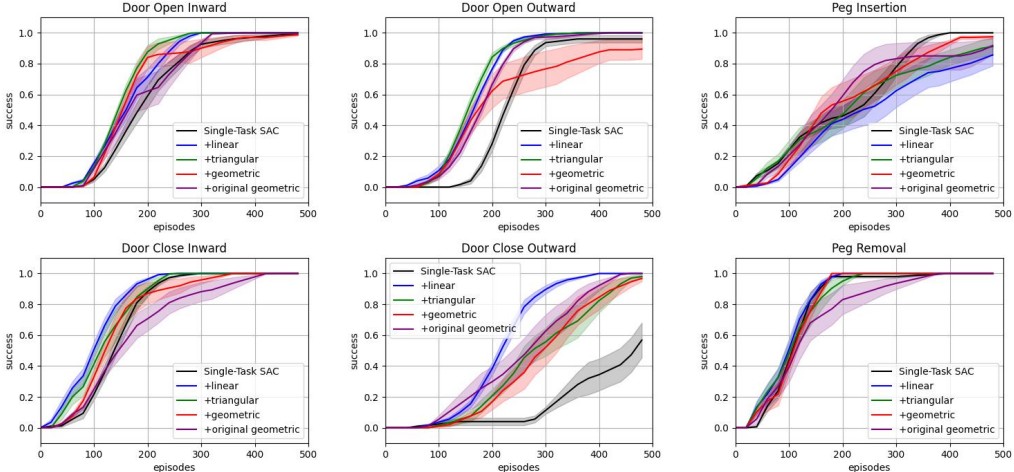

Figure 21: Evaluation curves of agent success rate using time reversal symmetry guided reward shaping with different potential types.

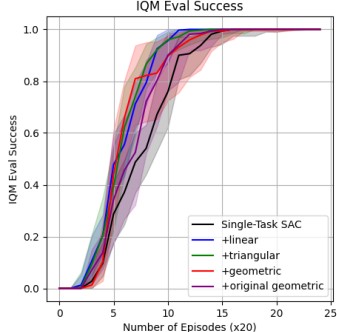

Figure 22: IQM of agent success rate using time reversal symmetry guided reward shaping with different potential types.

# H    Results of Time Reversal Symmetry Guided Reward Shaping in Robosuite

As shown in Figure 23, we plot the evaluation curves of time reversal symmetry guided reward shaping in 6 environments of robosuite, from which we can see clear performance gap between the baseline and using time reversal symmetry guided reward shaping.

# I    Results of Both Proposed Techniques in Robosuite

Full evaluation curves of combining trajectory reversal augmentation with dynamics-aware filtering and time reversal symmetry guided reward shaping are provided in Figure 24, which confirm that combining both techniques yields superior performance compared to using either component alone or the baseline method.

# J    Results of multi-task settings in Robosuite

Full evaluation curves of agent performance in 10 environments of Robosuite are shown in Figure 25.

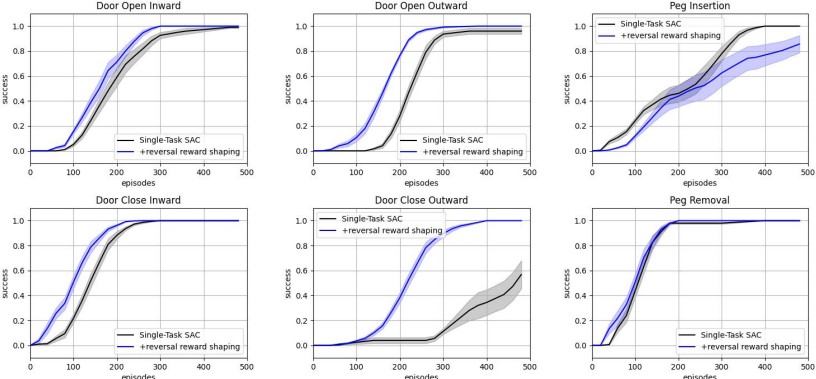

Figure 23: Evaluation curves of time reversal symmetry guided reward shaping in 6 environments of robosuite. "Single-Task SAC" serves as the baseline. "+reversal reward shaping" introduces time reversal symmetry guided reward shaping.

## K    Additional results of MT50 in Meta-World

Additional results of agent performance in both 12 reversible task pairs and all 50 environments of MT50 are shown in Figure 26, Table 2, Figure 27, and Figure 28.

## L    Compute Resources We Use

In all our experiments, we utilize a GPU server equipped with 8 cards that have either RTX-4090 or A6000 GPUs and are powered by AMD EPYC 7763 CPUs. For experiments in robosuite: training a single-task agent takes around 5 hours while training a multi-task agent for two tasks takes around 10 hours for 500 training episodes for each task. For experiments in MetaWorld: training a single-task agent takes around 2 hours while training a multi-task agent for two tasks takes around 4 hours for 500 training episodes for each task. For MT50, it takes around 3 days to train an agent that handles 50 tasks.

## M    Limitations

A key limitation of our work is the absence of real-robot experiments, as our current experiments are all conducted in simulation environments. While simulations enable efficient prototyping and scalability, they may oversimplify physical dynamics, sensor noise, or actuator constraints inherent in real-world robotic systems. Future work could address this gap by deploying the proposed method on physical robots, ensuring robustness and generalizability to practical applications. Like other data augmentation methods, our approach relies on prior knowledge of task structures—specifically, time reversal symmetry in our case.

## N    Broader Impacts

**Positive societal impacts**   : This work advances the sample efficiency of deep reinforcement learning (DRL) agents in robotics manipulation tasks, enabling faster and more cost-effective training for practical applications. By reducing the computational resources required for training, it lowers barriers to deploying robotic systems in real-world setting. Improved sample efficiency also minimizes energy consumption and hardware wear, aligning with sustainability goals. Furthermore, robust and efficient DRL methodologies can accelerate the development of autonomous systems that enhance productivity, safety, and accessibility, ultimately contributing to economic growth and societal well-being.

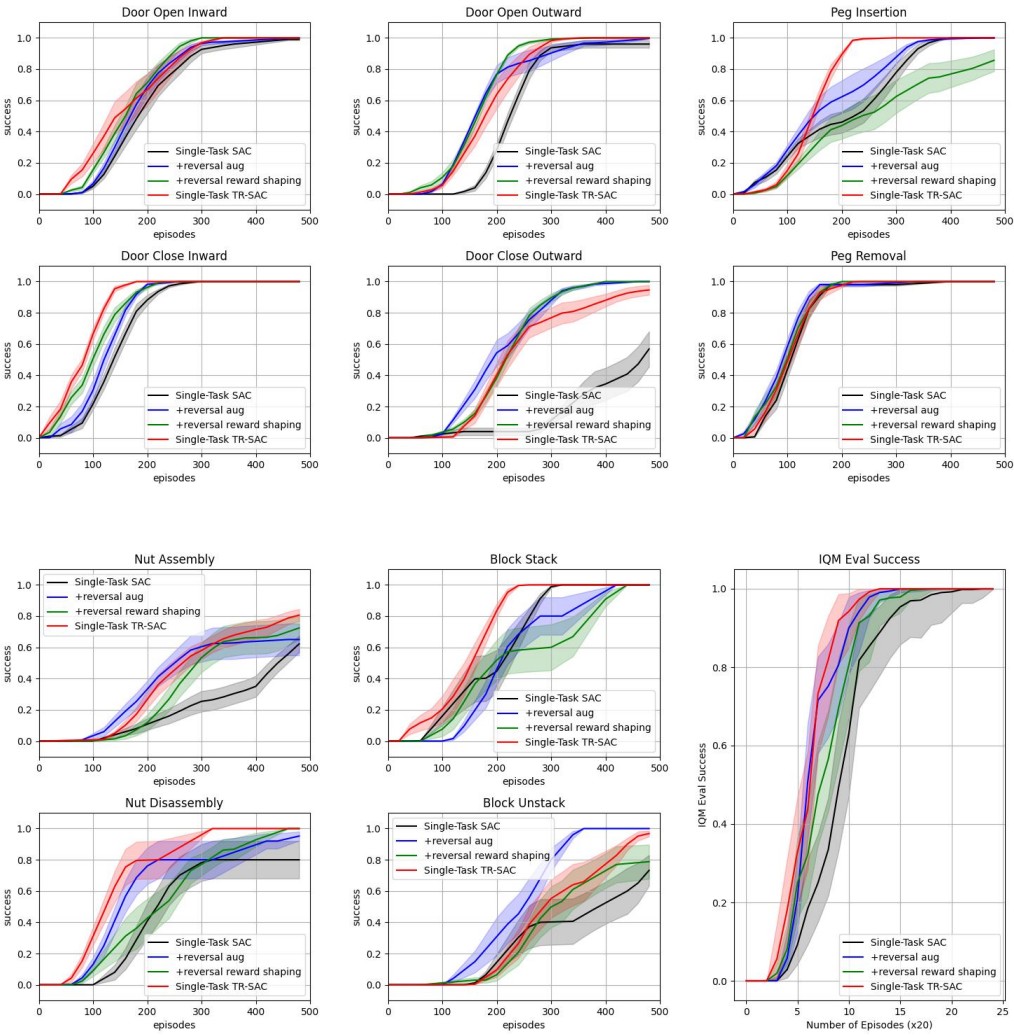

Figure 24: Evaluation curves of both components in 10 environments of Robosuite. "reversal aug" represents incorporating reversal augmentation with filtering. "reversal reward shaping" represents incorporating potential-based reward shaping. "Single-Task TR-SAC" represents our proposed method which combines both components.

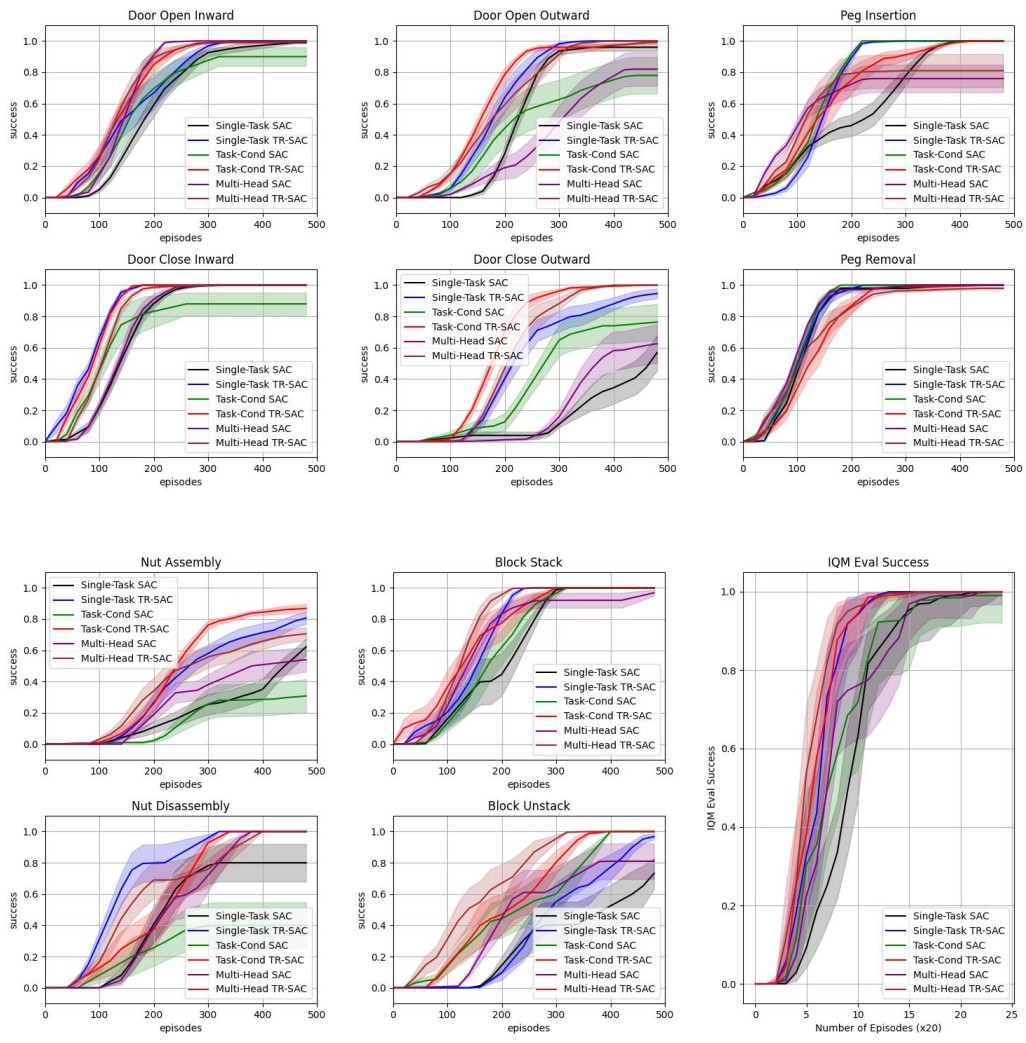

Figure 25: Evaluation curves for multi-task settings in 10 environments of Robosuite. "Task-Cond" and "Multi-Head" are short for "task-conditioned" and "multi-headed" respectively.

**Negative societal impacts**   To the best of our knowledge, we don't see any negative societal impacts of our work.

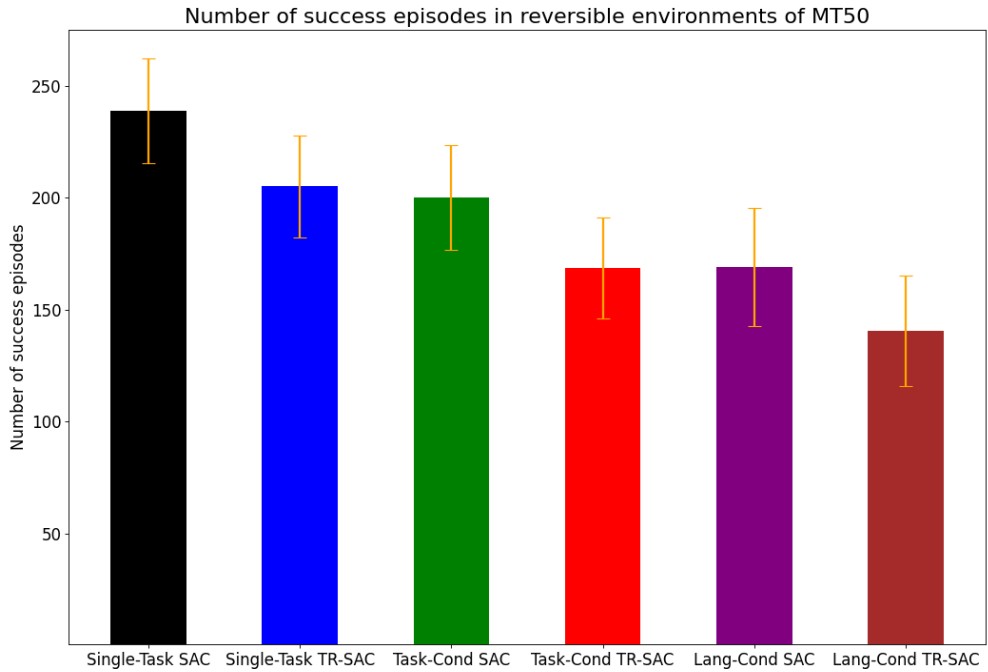

Figure 26: Average number of training episodes required to achieve a 100% success rate in 12 pairs of reversible environments of MT50..

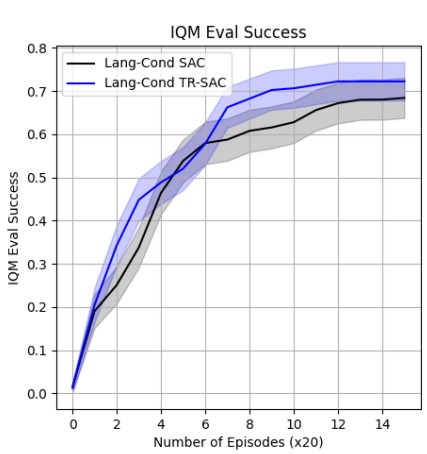

Figure 27: IQM for agent success rate in all 50 environments of MT50.

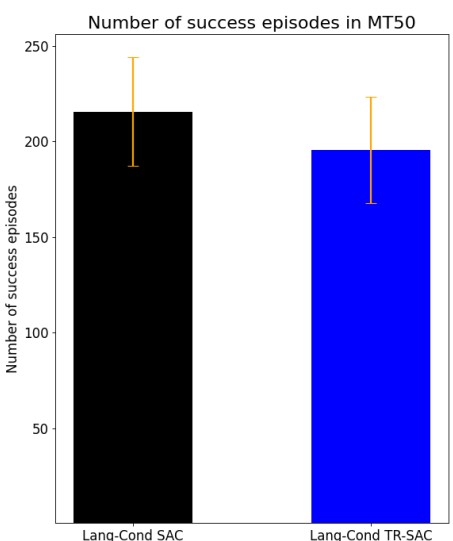

Figure 28: Average number of training episodes required to achieve a 100% success rate in all 50 environments of MT50.

Table 2: Number of training episodes required for 100% success rate for 12 pairs of reversible environments in MT50. Each value is averaged over five runs, with the mean and standard deviation reported. "Task-Cond" and "Lang-Cond" are short for "task-conditioned" and "language-conditioned" respectively. **Lower is better.**

| Environment | Single-Task SAC | Single-Task TR-SAC | Task-Cond SAC | Task-Cond TR-SAC | Lang-Cond SAC | Lang-Cond TR-SAC |
|---|---|---|---|---|---|---|
| assembly | 460±7 | 432±13 | 456±8 | 424±21 | 340±28 | **252±29** |
| disassemble | 500±0 | 500±0 | 500±0 | 500±0 | 424±21 | **364±24** |
| coffee pull | 444±16 | **408±16** | 460±4 | 468±9 | 416±24 | 416±24 |
| coffee push | 288±16 | 236±19 | 316±24 | 224±10 | 228±32 | **156±25** |
| door lock | 360±26 | 352±23 | 112±13 | 88±5 | 72±13 | **60±7** |
| door unlock | 196±22 | 148±14 | 144±7 | 88±9 | 96±10 | **72±5** |
| door open | 252±11 | 208±16 | 208±11 | **168±7** | 204±22 | 252±29 |
| door close | 100±4 | 88±1 | 80±2 | 48±1 | 52±4 | **44±5** |
| drawer open | 284±13 | 200±9 | 272±26 | 172±15 | 140±8 | **104±12** |
| drawer close | 40±3 | 16±1 | 16±2 | 12±1 | **8±1** | **8±1** |
| faucet open | 96±2 | 84±1 | 92±5 | 80±3 | 48±5 | **40±6** |
| faucet close | 124±6 | 156±7 | 72±2 | **60±3** | 76±2 | 64±5 |
| handle press | 32±4 | 20±2 | 20±2 | 24±3 | 24±1 | **20±0** |
| handle pull | 244±14 | 172±9 | 308±20 | **128±7** | 168±24 | 160±25 |
| peg insert side | 328±22 | 296±18 | 348±18 | **276±16** | 424±21 | 348±26 |
| peg unplug side | 164±13 | 128±9 | 76±3 | 68±3 | 180±24 | **32±1** |
| plate slide | 208±14 | 180±14 | 176±4 | 104±11 | 80±4 | **64±7** |
| plate slide back | 256±18 | 224±20 | 152±10 | 84±3 | 40±4 | **36±2** |
| plate slide side | 228±16 | 156±13 | 128±9 | 148±20 | 80±7 | **60±6** |
| plate slide back side | 196±12 | 148±4 | 140±7 | 152±8 | 104±9 | **72±5** |
| push | 288±17 | **140±10** | 272±24 | 244±18 | 404±27 | 328±30 |
| push back | 480±6 | 500±0 | **316±22** | 396±14 | 368±24 | 324±30 |
| window open | 84±1 | 64±1 | 76±5 | 52±2 | **24±1** | 44±5 |
| window close | 80±2 | 68±1 | 64±1 | **40±0** | 56±5 | 52±2 |
| **ALL** | 239±23 | 205±23 | 200±24 | 169±23 | 169±26 | **140±25** |

Table 3: Number of training episodes required for 100% success rate for all 50 environments in MT50. Each value is averaged over five runs, with the mean and standard deviation reported. "Lang-Cond" is short for "language-conditioned". **Lower is better.**

| Environment | Lang-Cond SAC | Lang-Cond TR-SAC | Environment | Lang-Cond SAC | Lang-Cond TR-SAC |
|---|---|---|---|---|---|
| assembly | 340±28 | **252±29** | sweep-into | 276±27 | **172±24** |
| disassemble | 424±21 | **364±24** | reach | **104±4** | 164±24 |
| coffee pull | 416±24 | 416±24 | reach wall | **128±11** | 188±22 |
| coffee push | 228±32 | **156±25** | stick-pull | 172±23 | **160±25** |
| door lock | 72±13 | **60±7** | sweep | 316±23 | **284±25** |
| door unlock | 96±10 | **72±5** | basketball | 500±0 | 500±0 |
| door open | **204±22** | 252±29 | bin picking | **348±26** | 360±24 |
| door close | 52±4 | **44±5** | box close | 352±27 | **296±25** |
| drawer open | 140±8 | **104±12** | coffee button | **52±3** | 64±7 |
| drawer close | **8±1** | **8±1** | button press | **28±2** | 36±2 |
| faucet open | 48±5 | **40±6** | button press wall | **76±2** | 76±6 |
| faucet close | 76±2 | **64±5** | button press topdown | 220±32 | **100±8** |
| handle press | 24±1 | **20±0** | button press topdown wall | 160±25 | **100±9** |
| handle pull | 168±24 | **160±25** | dial turn | 384±21 | **356±25** |
| handle pull side | **304±24** | 320±31 | handle press side | **32±3** | 36±2 |
| peg insert side | 424±21 | **348±26** | hammer | 172±23 | **160±25** |
| peg unplug side | 180±24 | **32±1** | hand insert | **236±31** | 256±28 |
| plate slide | 80±4 | **64±7** | lever pull | 280±27 | **252±30** |
| plate slide back | 40±4 | **36±2** | pick out of hole | **424±21** | 428±20 |
| plate slide side | 80±7 | **60±6** | pick place | 500±0 | 500±0 |
| plate slide back side | 104±9 | **72±5** | pick place wall | 416±24 | 416±24 |
| push | 404±27 | **328±30** | push wall | **344±28** | 356±25 |
| push back | 368±24 | **324±30** | shelf place | 428±20 | 428±20 |
| window open | **24±1** | 44±5 | soccer | 336±29 | **248±29** |
| window close | 56±5 | **52±2** | stick push | **140±9** | 156±6 |

| | Lang-Cond SAC | Lang-Cond TR-SAC |
|---|---|---|
| **ALL** | 216±28 | **196±28** |

