# OpenReview forum: "Time Reversal Symmetry for Efficient Robotic Manipulations in Deep Reinforcement Learning"
_NeurIPS.cc/2025/Conference — NeurIPS 2025 poster_

### Official Review · Reviewer_dzfv · 2025-06-30

**Clarity:** 4
**Significance:** 3
**Originality:** 3
**Rating:** 5
**Confidence:** 4

**Summary:**

The paper explores an intuitive approach to improving the sample efficiency of RL by using a time reversal symmetry. Time reversal symmetry is when an experience can be played backward in time in order to get information about how to perform the reversed version of a task, e.g. cloing a drawer versus opening a drawer. They explore two different methods. In the first (full time reversal), the method identifies those transitions that are fully reversible by training a forward and inverse dynamics model and evaluating the degree of match between them. Transitions that are fully reversible are reversed and added to the replay buffer. In the second method (partial time reveral), the portion of state that is reversible is used to guide RL exploration via a potential function that provides a dense reward signal. The paper presents compelling robosuite experiments that demonstrate that the method is helpful in most situations. Fig 6 is particularly compelling.

**Questions:**

It was all pretty clear to me.

**Ethical Concerns:**

["NO or VERY MINOR ethics concerns only"]

**Final Justification:**

I think this is a solid paper that should be accepted if possible. I like the intuition of the idea of time reversal symmetry. My main criticisms are: 1) I suspect there are more sophisticated ways of leveraging this intuition; 2) the current method is a little "heavy" in terms of requiring learning of additional forward and inverse models. Re the second criticism, as the reviewers point out in the rebuttal, this is probably not a huge ask since it is not uncommon to do this.

**Limitations:**

See my weaknesses above. I think complexity is a limitation. I also think that there there might be more effective ways to do this beyond data augmentation and dense rewardifying.

**Quality:**

3

**Strengths And Weaknesses:**

+ I liked the main idea of the paper. I liked that it was intuitive. This seems like an approach that should be helpful in a variety of situations.

+ I thought that the methods that were proposed were pretty reasonable and the experimental results indicate that they are generally helpful.

+ I thought the experiments were good. I was convinced.

- A key question here is whether this method passes the bang-for-buck threshold. In order to use this method, you need to learn forward and inverse dynamics models, significantly adding to the compute complexity of the alg. There is a benefit to doing so, but it may not be enough on its own to warrant learning those extra models. On the other hand, there are probably plenty of situations where folks are already learning those dynamics models and the method would definitely make sense there.

- I can't help but wonder if there might be more effective and sophisticated ways of leveraging the time reversal symmetry. There should be some systems for which time reversal should yield a 2x speedup. Are there ways to maybe incorporate time reveral directly into the neural model? Not clear, but interesting.

---

> ### Author Rebuttal · Authors · 2025-07-31
>
> We sincerely appreciate your positive evaluation, especially noting the intuitiveness of our approach.
> In response to your observations about weaknesses, below we provide answers in detail.
>
> **Weakness1**. "A key question here..."
>
> We agree that learning forward and inverse dynamics models has additional computational cost but emphasize that superior sample efficiency is a critical goal in deep RL.
> Like common techniques such as data augmentation or auxiliary losses, this added computation directly enables efficient learning with fewer samples and better final performance.
> Furthermore, our method seamlessly integrates with existing approaches that have already trained dynamics models, offering enhanced performance without incremental computational cost in such cases.
>
> **Weakness2**. "I can't help but wonder..."
>
> We appreciate your insightful suggestion about sophisticated ways of leveraging time reversal symmetry and we agree that it is an interesting topic.
> One possible approach is to incorporate the temporal symmetry into the neural network architecture design, similar to steerable CNNs handling spatial symmetry like rotation equivariance.
> Moreover, we suspect that the 2x speedup may be too optimistic.
> This would require idealized conditions rarely met in practice, including perfectly reversible dynamics and accurate backward policy execution.

---

### Official Review · Reviewer_Pawe · 2025-07-02

**Clarity:** 4
**Significance:** 2
**Originality:** 3
**Rating:** 5
**Confidence:** 4

**Summary:**

In this work, the authors present a simple yet effective trick for data augmentation in reinforcement learning methods using temporal symmetry that they call "Time Reversal Symmetry" augmentation. This augmentation take advantage of the following facts (i) often in a replay buffer, there exist reversible state transitions (ii) these reversible transitions can be identified with learned dynamics models (iii) once identified, they can be used to create a potential function to augment the reward. Once used properly, this method is able to increase the data efficiency of reinforcement learning methods, as shown in experiments in MetaWorld and RoboSuite environments.

**Questions:**

1. While the work only focuses on involutions created by forward and inverse dynamics models, could the authors think of some more interesting, maybe simpler involutions, which could take this beyond simple "negate the action sign" model? Perhaps even a learned action inverting model could give interesting results.
2. What would the results look like without the guiding demonstrations seeding the replay buffer? Would this method make a difference when exploring tabula rasa? If no, why not?

**Ethical Concerns:**

["NO or VERY MINOR ethics concerns only"]

**Final Justification:**

I appreciate the updates from the authors, and have therefore updated my confidence in my score. With some real robot experiments it would be a strong accept, but this work is valuable even without such experiments and justifies my "Accept" rating.

**Limitations:**

While the authors mention the lack of real world experiments, there are further limitations beyond that. Particularly, the method currently is limited to quasi-static environments, and would be hard to extend to domains such as video games or self-driving because the environments are generally dynamic and there is a definite "forward arrow of time" in such settings.

**Paper Formatting Concerns:**

Fonts on images can be small and hard to read on print. Please try to ensure the image fonts at least match the paper font size.

**Quality:**

3

**Strengths And Weaknesses:**

This work in particular is impressive in that it takes a simple idea, and with proper evaluation, shows that it leads to improvement in some real benchmark tasks. The idea is sensible, and the method design that leads around the idea feels like the "shortest path" needed to reach the effective method, which is desirable and yet rare in recent works. The ablation studies presented in the paper are also sensible, and does the ablations at the right level of abstraction to prove the necessity of each of the pieces.

While the idea behind the work and the presented method are strong, the benchmarking and the results leave something to be desired. Particularly, evaluating on two somewhat simple simulated environments with some demonstration data already in the buffer makes the tasks accomplished incredibly simple. At the same time, from the plots, it seems that the performance improvements are not always clear – often there are significant overlaps in the error bars, and some more rigorous evaluations with more seeds, or more thorough investigation on a larger set of simulated and real environments, while not critical at this stage, will make this work significantly stronger.

---

> ### Author Rebuttal · Authors · 2025-07-31
>
> Thank you for noting the quality and clarity of our work.
> We are happy that you recognize that the model design leading around our idea feels like the "shortest path".
> Below, we address each question thoroughly.
>
> **Q1**. "While the work..."
>
> We need to clarify that our method specifically learns an 'action inverting model' implemented via an inverse dynamics model.
> Unlike Barkley et al.'s approach [1] of simply negating actions in reversed transitions—which is not always valid—our model derives reverted actions through a learned inverse dynamics.
> However, for transitions without full time reversal symmetry, the inverse dynamics model may still generate invalid actions.
> To address this, we introduce a forward dynamics model to filter out such invalid transitions.
>
> **Q2**. "What would the results..."
>
> In the sparse-reward setting, exploring tabula rasa is a very challenging task for the RL agent.
> Consequently, agent performance without demonstrations drops substantially, which is a key reason that we do not include these experiments in the manuscript.
> On challenging tasks such as door opening/closing outward, nut assembly/disassembly, and block stacking/unstacking, the agents fail to make progress under sparse rewards alone.
> However, on other tasks like door opening/closing inward and peg insertion/removal, the IQM of success rate (shown in the tables below) reveal that the performance gap between TR-DRL and baseline methods widens significantly without demonstrations, confirming that time reversal symmetry provides guidance and partially compensates for missing demonstration data.
>
>  "Single-Task SAC": baseline; "+reversal aug": trajectory reversal augmentation with dynamics-aware filtering; "+reversal reward shaping": time reversal symmetry guided reward shaping.
> |Task|Number of Transitions |50k|100k|150k|200k|
> |:----------------- | :-------------------------------- | :------ | :------ | :------ | :------ |
> |**Door opening inward**||||||
> |Single-Task SAC|  | `0.00±0.00`  | `0.00±0.00`  | `0.00±0.00`  | `0.00±0.00`  |
> |+ reversal aug| | `0.00±0.00`  | `0.00±0.00`  | `0.00±0.00`  | `0.18±0.36`  |
> |+ reversal reward shaping || `0.00±0.00`  | **`0.13±0.26`** | **`0.31±0.41`** | `0.60±0.38`  |
> |Single-Task TR-SAC **(Ours)**|| `0.00±0.00`  | `0.11±0.22`  | `0.26±0.37`  | **`0.72±0.37`** |
> |**Door closing inward**||||||
> |Single-Task SAC|  | `0.00±0.00`  | `0.00±0.00`  | `0.05±0.09`  | `0.19±0.36`  |
> |+ reversal aug| | `0.00±0.00`  | `0.20±0.40`  | `0.20±0.40`  | `0.37±0.45`  |
> |+ reversal reward shaping || `0.03±0.06`  | **`0.29±0.28`** | `0.46±0.38` | `0.69±0.35`  |
> |Single-Task TR-SAC **(Ours)**|| `0.02±0.03`  | `0.29±0.35`  | **`0.55±0.41`**  | **`0.77±0.30`** |
> |**Peg Insertion**||||||
> |Single-Task SAC|  | `0.00±0.00`  | `0.00±0.00`  | `0.00±0.00`  | `0.23±0.24`  |
> |+ reversal aug| | `0.05±0.09`  | `0.37±0.33`  | `0.58±0.37`  | `0.95±0.07`  |
> |+ reversal reward shaping || `0.07±0.09`  | `0.37±0.44` | `0.55±0.45` | `0.75±0.38`  |
> |Single-Task TR-SAC **(Ours)**|| **`0.16±0.16`**  | **`0.50±0.39`**  | **`0.91±0.07`**  | **`1.00±0.00`** |
> |**Peg Removal**||||||
> |Single-Task SAC|  | `0.00±0.00`  | `0.09±0.17`  | `0.27±0.39`  | `0.64±0.37`  |
> |+ reversal aug| | **`0.25±0.26`**  | `0.81±0.31`  | **`1.00±0.00`**  | **`1.00±0.00`**  |
> |+ reversal reward shaping || `0.00±0.00`  | `0.23±0.39` | `0.52±0.44` | `0.63±0.45`  |
> |Single-Task TR-SAC **(Ours)**|| `0.08±0.10`  | **`0.86±0.15`**  | **`1.00±0.00`**  | **`1.00±0.00`** |
>
> [1]. Barkley et al. "An Investigation of Time Reversal Symmetry in Reinforcement Learning." https://arxiv.org/abs/2311.17008

---

> > ### Comment · Reviewer_Pawe · 2025-08-03
> > **Thank you for the updates.**
> >
> > Thank you for your clarification and list of updates. I have updated my confidence in this work accordingly. As other reviewers have mentioned, only thing that can make this paper stronger is having real robot experiments.

---

> ### Author Response · Authors · 2025-08-04
> **Thanks for your reply**
>
> Dear Reviewer Pawe,
>
> We appreciate your positive feedback and increased confidence in our paper. Thank you again for your reply.
>
> Best, Authors

---

### Official Review · Reviewer_TdWt · 2025-07-03

**Clarity:** 2
**Significance:** 2
**Originality:** 2
**Rating:** 5
**Confidence:** 4

**Summary:**

This paper introduces Time Reversal Deep RL (TD-DRL), a framework for exploring time-reversal symmetry in robotics tasks to improve the data efficiency of RL. TR-DRL generates additional augmented data that respect the dynamics of the task by reversing trajectories, and then incorporates this data into off-policy RL algorithms. Empirically, TR-DRL improves data efficiency on Robosuite and Metaworld tasks.

**Questions:**

1. "most reversed transitions are filtered out by the dynamics-consistent filter" Could you support this empirically? It'd be interesting to see  how much is filtered out.
1. "The training process alternates between two reversible tasks." Is this a critical part of the algorithm, or was this just chosen for convenience?
1. Do the authors have any sense of how the augmented data compares to additional policy-generated data? In particular, suppose we replaced the augmented data with additional data collected by the agent (and suppose we give the agent this data "for free" in that we don't count this data as additional timesteps).

**Ethical Concerns:**

["NO or VERY MINOR ethics concerns only"]

**Final Justification:**

The paper is properly motivated, technically correct, and adequately supports stated claims. My comments were mostly clarification questions and can easily be addressed by minor changes to a camera-ready version.

I do want to emphasize that because the method is of most interest to roboticists but is not evaluated on physical robotics tasks, impact is limited. However, I put much more weight in the "correctness" and "relevance" aspects of research, so I do no seet this drawback as a reason to reject given the work's other merits.

**Limitations:**

Yes, but I think it's worth mentioning that this method assumes we know the task has time reversibility. I don't think this is a significant shortcoming though, since all data augmentation papers assume knowledge of some function that can transform observed data into additional training data (e.g. an augmentation function). This is just an inherent limitation of data augmentation methods. Corrado & Hanna [1] and Pitis et al. [2] discuss this point.

1. Corrado & Hanna. "Understanding when Dynamics-Invariant Data Augmentations Benefit Model-free Reinforcement Learning Updates." https://arxiv.org/abs/2310.17786
3. Pitis et al. "Counterfactual Data Augmentation using Locally Factored Dynamics."  https://arxiv.org/abs/2007.02863

**Quality:**

3

**Strengths And Weaknesses:**

# Strengths

1. Partial time reversal symmetry appears in many real world robotics, so the topic is relevant and practically useful.
2. TR-DRL outperforms baselines by a fair margin in all tasks consider.
3. From the ablation studies, I gather that TR-DRL is fairly robust to hyperparameter choices and that each component of the method helps improve performance.

# Weaknesses

1. **Baselines.** TR-DRL is the only method that leverages this knowledge and thus naturally has an advantage over vanilla baselines. I wouldn't say this is necessarily an unfair advantage, though; a core contribution of this paper is showing the benefit of incorporating knowledge of time reversal into training. However, I think some discussion is warranted on other ways we could potentially leverage this information. e.g. learning reversible action representations [1].
To rephrase my comment: Experiments show that adding time-reversed augmentated data to the replay buffer improves data efficiency. Are there other ways to integrate time-reversal information? How do they compare?

1. **Time reversal examples are a bit unclear.** My understanding other partial time reversal is rooted in my understanding of locally factored dynamics discussed by Pitis et. al. [2] (see the next section below for additional context). I had a bit of trouble understanding what Example 2 was talking about until I made this connection. Are we saying that the door state is reversible because there exists some action that can reverse the door's trajectory?

1. **Mild impact.** The paper addresses a very specific problem that would be primarily of interest to robotics-focused practitioners rather than the broader RL community.

# Other comments

1. The definition of partial time reversibility (line 149) reminds me of the definition of "locally factored dynamics" discussed in CoDA by Pitis et. al. [2]. Locally factored dynamics means that after taking an action, only a small number of components of the next state s’ change. For instance, an object won't move unless the agent touches it, so if the agent and object are physically separated, the agent's dynamics are completely independent of the object's dynamics. This local structure allows Pitis et al. [2] to generate counterfactual transitions by swapping parts of next states across trajectories that share common context in the unaffected dimensions. You may be able to leverage ideas/definitions from [2] to motivate partial time reversibility. Followup works MoCoDA [3] and RoCoDA [4] may be of interest.

# Minor Comments

1. "existing work (see Related Work in Section 2) predominantly focuses on spatial symmetries, 28 such as translation, reflection, and rotation..." The authors might also mention augmentation methods that aim to improve robustness to noise (i.e. essentially all visual augmentation works, S4RL [5], and "sample enhancement" in [6])
2. Line 33: "speed of action execution" in this context, speed refers to the frequency at which actions are selected, right?
3. Figure 2: I should think of the reversal of this trajectories as reading the images in reverse from right to left, correct? Also, I suggest writing "Figure 2: Examples of reversible and irreversible trajectories" would help clarify what this figure is showing. (a) Reversible: ... (b) Irreversible ..." The juxtaposition of (a) and (b) made me think (b) was meant to represent the reversal of (a).
4. Line 105: the definitions of the dynamics function suggests stochastic dynamics, but does stochasticity interfere with reversibility?
1. Line 111: The overhead arrow notation is not explicitly defined prior ot using it.
1. Line 115: I think it's worth emphasizes somewhere that these sorts of symmetries frequently arise in the real world, so it's reasonable to assume these involutions exist.
1. When defining inverse dynamics models, it would help to unify notation a bit. For instance, $f_a, f_s, $ could denote dynamics for
1. Figures 5 and 6 would be much clearer if the results for each algorithm pair (TR vs. no TR)were plotted separately.
1. Line 115 "iin" typo
---
# References

1. Xie et al. "Learning Latent Representations to Influence Multi-Agent Interaction." https://arxiv.org/abs/2011.06619
3. Pitis et al. "Counterfactual Data Augmentation using Locally Factored Dynamics."  https://arxiv.org/abs/2007.02863
4. Pitis et al. MoCoDA: Model-based Counterfactual Data Augmentation. https://arxiv.org/abs/2007.02863https://arxiv.org/abs/2210.11287
4. Ameperosa et al. "RoCoDA: Counterfactual Data Augmentation for Data-Efficient Robot Learning from Demonstrations." https://arxiv.org/abs/2411.16959
1. Sinha et al. S4RL: Surprisingly Simple Self-Supervision for Offline Reinforcement Learning. https://arxiv.org/abs/2103.06326
2. Qiao et al. "Efficient Differentiable Simulation of Articulated Bodies." https://arxiv.org/abs/2109.07719

---

> ### Author Rebuttal · Authors · 2025-07-31
>
> Thank you for summarizing our work and providing thoughtful feedback that helps us strengthen both the presentation and technical depth of our paper.
> Below we answer your questions in detail.
>
> **Weakness1**. "Baselines..."
>
> As detailed in related works (Section 2), there are several other ways to integrate time-reversal information.
>
> Some existing works exploit reversibility from goal states to enhance exploration of the agents, similar to what we would like to achieve through the time reversal guided reward shaping technique.
> Our method differs from them in that our proposed trajectory reversal augmentation technique leverages time reversal symmetry not only from goal states but globally for every transition in the trajectory, enabling a broader application of time symmetry across the entire state space.
>
> Other prior works are designed to avoid irreversible actions of the agent for safer exploration.
> One example is "learning a reset policy alongside the normal policy to prevent agents from entering non-reversible states, ensuring safety in exploration phase and achieving better training efficiency".
> Unlike their reset policy that sets the initial state as the ending state of the current policy and the goal state as the task's starting point, our method treats two reversible tasks independently, with initial and goal states defined separately for each task.
> Moreover, our method is orthogonal to theirs and can be integrated to enhance the training of their reset policy.
>
> **Weakness2**. "Time reversal examples..."
>
> We have updated our manuscript to recall state decomposition proposed by this work when we introduce the example of partial time reversal symmetry.
> The answer to your question in **Weakness2** is: Yes.
> By our definition in Section 4.1, partial time reversal symmetry indicates that a subset of the whole state is reversible.
> In the example of door opening/closing inward, it means that the door's trajectory can be reversed by some action.
>
> **Minor1**. "existing work..."
>
> Thank you for highlighting this important connection.
> We have expanded the related work to explicitly include how data augmentation methods improve noise robustness and cited these two works as key examples.
>
> **Minor2**. "Line 33:..."
>
> Yes, "the speed of action execution" is the frequency at which actions are selected.
> This is often controlled by a hyperparameter called "action repeat" in RL methods.
> A larger value of "action repeat" indicates a lower control frequency.
>
> **Minor3**. "Figure 2:..."
>
> Thanks for this constructive suggestion.
> We have updated our manuscript accordingly.
>
> **Minor4**. "Line 105:..."
>
> As defined in line 110, time reversal symmetry applies to probability distributions over transitions.
> Consequently, stochastic dynamics does not interfere with reversibility, as it can remain statistically consistent under time reversal.
>
> **Minor5**. "Line 111:..."
>
> To avoid confusion, we have explicitly defined the overhead arrow notation as time reversal operation on $\mathcal{S}$ or $\mathcal{A}$ at line 110 of our manuscript.
>
> **Minor6**. "Line 115:..."
>
> Thank you for highlighting this important point.
> As noted by Barkley et al. (cited at line 110), such time symmetries commonly emerge in physical systems.
> We have now emphasized this point explicitly near line 115 in our manuscript.
>
> **Minor7**. "When defining..."
>
> Both $f_{\mathcal S}$ and $f_{\mathcal A}$ are used in Section 3 for involutions over states and actions.
> We choose to stick with the current notations for inverse and forward dynamics models ($h$ and $g$) to avoid misunderstanding.
>
> **Minor8**. "Figures 5 and 6..."
>
> Due to page limit in the main text, we combine the results of all methods into a single plot.
> We will include separate plots for each algorithm pair (TR vs. no TR) in the appendix for better readability.
>
> **Question1**. "most reversed transitions..."
>
> We empirically validate this point through quantitative analysis.
> We have tested our filtering method on the agent's saved replay buffer, which confirms that only 27.27\% of transitions in door closing inward task pass filtering versus 63.13\% for door closing outward task.
> Specifically, when closing a door inward, transitions of the gripper approaching the door can be reverted, while the transitions of the gripper pushing the door fail to pass filtering due to their violations of learned environment dynamics.
>
> **Question2**. "The training process alternates..."
>
> We would like to clarify that data collection (agent interacting with environments) in our method alternates between two tasks purely for implementation convenience.
> It is expected to be more efficient to interact with the two environments in parallel.
> During agent update step, we sample transitions from replay buffers of both environments simultaneously, and update all models with these transitions.
>
> **Question3**. "Do the authors..."
>
> Time reversal augmented data differs from policy generated data in quality and utility.
> One advantage of policy generated data is that it is 100\% realistic and obeys environment dynamics, while time reversal augmented data, even after filtering, may not be fully dynamics-consistent.
> Conversely, there are two advantages of time reversal augmented data.
> First, it could help improving agent exploration at the beginning of the training, since the trajectories from two tasks cover distinct regions of the state space.
> This synergy is validated by the results shown in Figure 22 of our manuscript, where doubling the amount of policy generated data can not reach the performance of time reversal augmented data in this case.
> Second, the agent may converge at different speeds on the two tasks.
> A successful trajectory from one task can effectively guide the training for the other.
>
> **Limitations**. "Yes, but I think it's worth mentioning..."
>
> We fully agree with this point and have explicitly included the following sentence in the limitations section of our manuscript.
> "Like other data augmentation methods, our approach relies on prior knowledge of task structures—specifically, time reversal symmetry in our case."

---

> > ### Comment · Reviewer_TdWt · 2025-08-02
> >
> > Thank you for the detailed response! I feel that my comments were appropriately addressed; they were mostly clarification questions and can easily be addressed by minor changes to a camera-ready version. I will raise my score to accept, since the paper is properly motivated, technically correct, and adequately supports stated claims.
> >
> > I do still think that the paper in its current state has mild impact; the method is of most interest to roboticists, though experiments are all simulated. Showing improved data efficiency in physical robotics tasks would emphasize the impact. Having said that, I put much more weight in the "correctness" and "relevance" aspects of research, and am overall happy with the work.

---

> > > ### Author Response · Authors · 2025-08-03
> > >
> > > Dear Reviewer TdWt,
> > >
> > > We appreciate your timely response! Thank you again for your constructive suggestions for revising our paper.
> > >
> > > Best, Authors

---

### Official Review · Reviewer_fKuw · 2025-07-03

**Clarity:** 3
**Significance:** 3
**Originality:** 3
**Rating:** 4
**Confidence:** 4

**Summary:**

The paper presents Time Reversal Symmetry Enhanced Deep Reinforcement Learning to address challenges in robotic manipulation tasks within DRL. While previous methods primarily leverage spatial symmetries, this paper focuses on exploiting time reversal symmetry, considering that certain robot manipulations can be reversed in time, such as opening and closing a door or inserting and removing a peg. The results suggest that TR-DRL significantly enhances learning efficiency and final performance compared to baseline methods.

**Questions:**

1. Can the time reversal symmetry techniques be applied to a wider range of robotic tasks, or are they limited to tasks with explicitly reversible actions (e.g., peg insertion/removal)? It would be helpful if the authors could provide insight into how broadly these methods could be applied.

2. Given that the paper does not include real-robot evaluations, how confident are the authors that their approach will perform well in a physical robot, particularly in environments with noise or variability in actuator performance? Future work could explore bridging this gap between simulation and real-world testing.

3. In partially reversible tasks, how does the agent handle situations where only certain components of the state are reversible (e.g., door state vs. robot state)? A more detailed explanation of how partial reversibility impacts training would be useful.

**Ethical Concerns:**

["NO or VERY MINOR ethics concerns only"]

**Final Justification:**

Resolved via rebuttal: The authors clarified scope and mechanics: distinction between FTR and PTR, the role of the dynamics-consistent filter, and why potential-based reward shaping is used to exploit partial reversibility without contaminating data. They also positioned the contribution as primarily improving the simulation training phase, which is reasonable.

Unresolved: No real-robot evaluation or concrete sim-to-real strategy; limited robustness analysis under sensor/actuator noise and inverse-dynamics model errors; unclear generalization beyond tasks with clear time-reversal structure; added engineering complexity (inverse dynamics + filtering) not fully quantified.

Camera-ready requests:

1. Include at least a pilot real-robot or a strong sim-to-real study with domain randomization/noise ablations;

2. Provide compute/complexity breakdown and sensitivity to inverse-dynamics inaccuracies;

3. Broaden coverage of partially reversible tasks with diagnostics/guidelines for identifying TR symmetry.

**Limitations:**

See above.

**Quality:**

3

**Strengths And Weaknesses:**

Strengths:

1. The introduction of time reversal symmetry in DRL is a novel and potentially impactful contribution. While spatial symmetries have been explored extensively, time reversal symmetry has not been fully leveraged in DRL.

2. The two proposed techniques—trajectory reversal augmentation and time reversal guided reward shaping—are well thought out and methodically sound. The dynamics-consistent filter for ensuring valid reversed transitions is an excellent addition.

Weaknesses:

1. **Lack of Real-World Experiments**: While the paper provides extensive simulation-based experiments, the lack of real-world robotic evaluations is a key limitation. This is particularly relevant in robotics, where simulations often fail to capture real-world complexities such as sensor noise, physical dynamics, and actuator constraints.

2. **Generalization to Other Domains**: The proposed methods are shown to work in robotics, but it is unclear how easily these techniques might generalize to other domains, such as those that do not have clear time reversal symmetry.

3. **Complexity**: While the proposed methods are valuable, they introduce additional complexity into the reinforcement learning setup, such as the need for inverse dynamics models and filtering. The practical implementation of these components could be challenging in some settings.

---

> ### Author Rebuttal · Authors · 2025-07-31
>
> Thank you for indicating our contributions and providing thoughtful feedback on our work.
> In this work, we focus on robotics tasks with time reversal symmetry, and evaluate our proposed method in simulation.
> We definitely agree that applying our approach in a real-world setting is an important step to prove the effectiveness of our method for solving those time-symmetric robotics tasks.
> And it is an interesting direction to learn the temporal symmetry if it does not obviously exist in tasks.
> We leave the real-world experiments and learning temporal symmetry as future work.
> Below, we answer each question in detail.
>
> **Q1**. "Can the time reversal..."
>
> We would like to clarify that our method exploits two types of symmetry: full time reversal symmetry (FTR) and partial time reversal symmetry (PTR).
> Peg insertion/removal and door opening/closing inward are examples of FTR and PTR task pairs respectively.
> In door opening/closing inward, most reversed transitions are filtered out by the dynamics-consistent filter, since the agent cannot reverse the action (i.e., from "push the door" to "grasp the handle and pull the door") without first grasping the handle.
> The detailed examples of partial reversibility can be found in Example 2 of Section 4.1.
> We design a time reversal reward shaping mechanism to handle these partially reversible transitions.
> A potential model is trained with successful trajectories of task A and used to guide the agent training of task B through potential-based reward shaping, as detailed in Section 4.3.
>
> **Q2**. "Given that the paper..."
>
> As stated at the beginning of the response, we admit the significance of real-world experiments and leave it as future work.
> Considering the difficulties of conducting RL directly under real-world setting, a common practice is to train the policy with RL in simulation and deploy the policy in real-world by solving the sim-to-real gap.
> Our approach can effectively facilitate the learning process during the simulation phase.
>
> **Q3**. "In partially reversible tasks..."
>
> For tasks where only subsets of states exhibit time reversal symmetry, reversed trajectories obtained from trajectory reversal augmentation often violate true environment dynamics, leading to low quality data that may even hurt agent training.
> Instead, our method leverages time reversal guided reward shaping to exploit the partial reversibility.
> Intuitively, for a high-reward trajectory from one task (door from open to closed), if a trajectory from another task can achieve the reversed object states (door from closed to open), it should likewise receive a high reward.
> This motivates us to employ reward shaping to guide the RL agent with these partially reversible transitions, as detailed in Section 4.3.
> Furthermore, we choose to use potential-based reward shaping as it preserves policy optimality and train a potential model so that we can fully utilize all successful trajectories.

---

> > ### Comment · Reviewer_fKuw · 2025-08-06
> >
> > Since the limitations are not addressed, I will keep my score of 4.

---

> > > ### Author Response · Authors · 2025-08-06
> > > **Thanks for your reply**
> > >
> > > Dear Reviewer fKuw,
> > >
> > > We will implement our method and apply the idea of time reversal symmetry in real-world scenarios. Thank you again for your reply.
> > >
> > > Best, Authors

---

### Note · Authors · 2025-08-16

We would like to thank all the reviewers for their efforts in reviewing our paper and providing us with helpful feedback.

Our work explores how to exploit temporal symmetry in deep RL, an underexplored direction compared to spatial symmetry.
The reviewers noted several positive aspects of our submission.
Time reversal symmetry occurs frequently in real-world robotics, showcasing its practical relevance (Reviewer TdWt).
Our main contribution is to show that our proposed techniques with time reversal symmetry are effective in improving data efficiency of deep RL.
Our two proposed techniques, trajectory reversal augmentation and time-reversal guided reward shaping, are technically sound (Reviewer fKuw).
Simultaneously, our method is described as the "shortest path" solution to an effective method (Reviewer Pawe) and praised for its intuitiveness and broad applicability (Reviewer dzfv).
The results in simulation environments are compelling (Reviewer dzfv), demonstrating robustness to hyperparameter choices and consistent performance gains from each proposed technique (Reviewer TdWt).
Crucially, our ablation studies precisely isolate each proposed technique and prove the necessity of each component (Reviewer Pawe).

During the rebuttal process, we have thoroughly answered the questions raised by each reviewer.
Specifically, we have included additional experimental results requested by Reviewer TdWt (Question1) and Reviewer Pawe (Question2), strengthening our validation.
Also, we have improved the writing of our paper for better readability and logic flow, thanks to detailed suggestions proposed by reviewer TdWt.

Finally, we agree that real-world experimental validation would be important.
However, our work is a first step to demonstrate how to effectively exploit time reversal symmetry in deep RL methods.
We leave this real-world validation to future work.

---

### Decision · Program_Chairs · 2025-09-17

**Decision:**

Accept (poster)

**Comment:**

The paper is in the line of research where symmetries in robot motions are used to make robot manipulation more efficient when using reinforcement learning for optimization. This paper focuses on time reversal symmetries where the idea is that for example a robot can sometimes reverse its actions, for example, walk forwards, and then walk back backwards. Previously, other kinds of symmetries such as symmetric shapes of objects have been used.

A strength of the paper is the introduction of the time reversal symmetries to robot reinforcement learning which is a strong contribution. While the contribution is at the moment limited to the robotics setting, it is of sufficient impact.

A weakness of the paper is lack of real robot experiments, or, sim2sim experiments with noise or other kind of perturbations to the environment. However, even without real robot experiments the paper provides an interesting contribution.

During the rebuttal the authors added further experimental results and answered satisfactorily reviewer questions. Also the applicability to other domains was discussed. This was left out of the scope of the paper.